# Absolute Zero:
# Reinforced Self-play Reasoning with Zero Data

**Andrew Zhao**[1]   **Yiran Wu**[3]   **Yang Yue**[1]   **Tong Wu**[2]   **Quentin Xu**[1]   **Matthieu Lin**[1]
**Shenzhi Wang**[1]   **Qingyun Wu**[3]   **Zilong Zheng**[2]*   **Gao Huang**[1]*
[1]Tsinghua University    [2]BIGAI    [3]Penn State University
zqc21@mails.tsinghua.edu.cn   yiran.wu@psu.edu
zlzheng@bigai.ai   gaohuang@tsinghua.edu.cn

## Abstract

Reinforcement learning with verifiable rewards (RLVR) has shown promise in enhancing the reasoning capabilities of large language models by learning directly from rule-based outcome rewards. Recent RLVR works that operate under the *zero setting* avoid supervision in labeling the reasoning process, but still depend on manually curated collections of questions and answers for training. The scarcity of high-quality, human-produced examples raises concerns about the long-term scalability of relying on human supervision, a challenge already evident in the domain of language model pretraining. Furthermore, in a hypothetical future where AI surpasses human intelligence, tasks provided by humans may offer limited learning potential for a superintelligent system. To address these concerns, we propose a new RLVR paradigm called *Absolute Zero*, in which a single model learns to propose tasks that maximize its own learning progress and improves reasoning by solving them, without relying on any external human or distillation data. Under this paradigm, we introduce the Absolute Zero Reasoner (AZR), a system that self-evolves its training curriculum and reasoning ability. AZR uses a code executor to both validate self-proposed code reasoning tasks and verify answers, serving as an unified source of verifiable feedback to guide open-ended yet grounded learning. Despite being trained entirely *without external data*, AZR achieves overall SOTA performance on coding and mathematical reasoning tasks, *outperforming existing zero-setting models* that rely on tens of thousands of *in-domain human-curated examples*. Furthermore, we demonstrate that AZR can be effectively applied across different model scales and is compatible with various model classes.

## 1 Introduction

Large language models (LLMs) have recently achieved remarkable improvements in reasoning capabilities by employing Reinforcement Learning with Verifiable Rewards (RLVR) [37]. Unlike methods that explicitly imitate intermediate reasoning steps, RLVR uses only outcome-based feedback, enabling large-scale reinforcement learning over vast task datasets [22, 75, 33, 52, 51, 81]. A particularly compelling variant is the *"zero"* RLVR paradigm [22], which forgoes any cold-start distillation data, using neither human-generated nor AI-generated reasoning traces, and applies RLVR directly on the base model with task rewards. However, these methods still depend heavily on expertly curated distributions of reasoning question–answer pairs, which raises serious concerns about their long-term scalability [76]. As reasoning models continue to advance, the effort required to construct large-scale, high-quality datasets may soon become unsustainable [97]. A similar scalability bottleneck has already been identified in the domain of LLM pretraining [73]. Furthermore, as AI

---

*Corresponding author.

39th Conference on Neural Information Processing Systems (NeurIPS 2025).

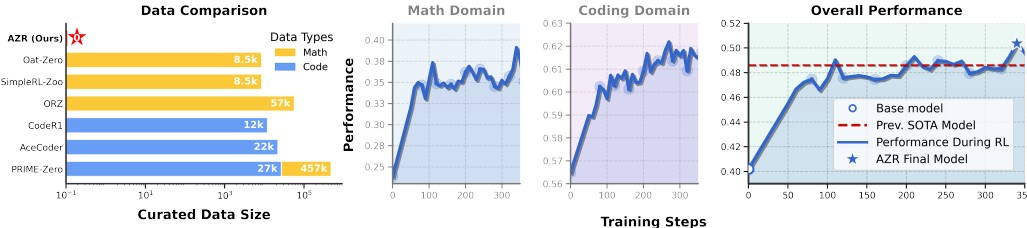

Figure 1: **Absolute Zero Reasoner (AZR) achieves state-of-the-art performance with ZERO DATA**. Without relying on any gold labels or human-defined queries, Absolute Zero Reasoner trained using our proposed self-play approach demonstrates impressive general reasoning capabilities improvements in both math and coding, despite operating entirely out-of-distribution. Remarkably, AZR surpasses models trained on tens of thousands of expert-labeled in-domain examples in the combined average score across both domains, and also reaches SOTA in the coding domain.

systems continue to evolve and potentially exceed human intellect, an exclusive dependence on human-designed tasks risks imposing constraints on their capacity for autonomous transcendence [31]. This underscores the need for a new paradigm that begins to explore possibilities beyond the constraints of human-designed tasks and prepares for a future in which AI systems may surpass human intelligence.

To this end, we propose *"Absolute Zero"*, a new paradigm for reasoning models in which the model simultaneously learns to define tasks that maximize learnability and to solve them effectively, enabling self-evolution through self-play without relying on external data. In contrast to prior self-play methods that are limited to narrow domains, fixed functionalities, or learned reward models that are prone to hacking [68, 5, 7], the *Absolute Zero* paradigm is designed to operate in open-ended settings while remaining grounded in a real environment. It relies on feedback from the environment as a verifiable source of reward, mirroring how humans learn and reason through interaction with the world, and helps prevent issues such as hacking with neural reward models [31]. Similar to AlphaZero [68], which improves through self-play, our proposed paradigm requires no human supervision and learns entirely through self-interaction. We believe the Absolute Zero paradigm represents a promising step toward enabling large language models to autonomously achieve superhuman reasoning capabilities.

Building on this new reasoning paradigm, we introduce the *Absolute Zero Reasoner (AZR)*, which proposes and solves code reasoning tasks. We cast code executor as an open-ended yet grounded environment, sufficient to both validate task integrity and also provide verifiable feedback for stable training. We let AZR construct tasks that require reasoning and inference about a specific element in a program, input, or output triplet, corresponding to three complementary modes of reasoning: induction, abduction, and deduction. We train the entire system end-to-end with a newly proposed reinforcement learning advantage estimator tailored to the multitask nature of the proposed approach.

Despite being trained entirely without any in-distribution data, AZR demonstrates remarkable capabilities across diverse general reasoning tasks in math and coding. In mathematics, AZR achieves competitive performance compared to zero reasoner models explicitly fine-tuned with domain-specific supervision. In coding tasks, AZR establishes a new state-of-the-art performance, surpassing models specifically trained with curated code datasets using RLVR. Furthermore, AZR **outperforms all previous models** by an average of 1.8 absolute points compared to models trained in the "zero" setting using in-domain data. These surprising results highlight that general reasoning skills can emerge without human-curated domain targeted data, positioning Absolute Zero as an promising research direction and AZR as a first effective instantiation. See Section D.1 for more interesting findings.

## 2 The Absolute Zero Paradigm

### 2.1 Preliminaries

**Supervised Fine-Tuning (SFT).** SFT requires the datasets of task-rationale-answer demonstrations $\mathcal{D} = \{(x, c^\star, y^\star)\}$, where $x$ is the query, $c^\star$ is the gold chain-of-thought (CoT) and $y^\star$ is the gold answer, all provided by human experts or superior AI models. The model trains to imitate the reference responses to minimize the conditional negative log-likelihood [55]:

$$\mathcal{L}_{\mathrm{SFT}}(\theta) \;=\; -\,\mathbb{E}_{(x,c^\star,y^\star)\sim\mathcal{D}} \log \pi_\theta \left(c^\star, y^\star \mid x\right). \tag{1}$$

At the frontier level, the absence of stronger models for distillation and the poor scalability of expert human labeling have led researchers to explore RL as a means to enhance model reasoning.

**Reinforcement Learning with Verifiable Rewards (RLVR).** To move beyond the limits of pure imitation, RLVR only requires a dataset of task and answer $\mathcal{D} = \{(x, y^\star)\}$, without labeled rationale. RLVR allows the model to generate its own CoT and calculate a verifiable reward with the golden answer $r(y, y^\star)$. However, the learning task distribution $\mathcal{D}$, with its set of queries and gold answers are still labeled by human experts. The trainable policy $\pi_\theta$ is optimized to maximize expected reward:

$$J_{\mathrm{RLVR}}(\theta) \;=\; \mathbb{E}_{(x,y^\star)\sim\mathcal{D},\ (c,y)\sim\pi_\theta(\cdot\,\mid x)}\left[r(y, y^\star)\right]. \tag{2}$$

In summary, both SFT and RLVR still rely on human-curated datasets of either queries, demonstrations, or answers, which limit scalability. The Absolute Zero paradigm removes this dependency by allowing the model to generate, solve, and learn from its own interactions with the environment by self-play.

## 2.2 Absolute Zero

We propose the Absolute Zero (AZ) paradigm, where during training, the model simultaneously proposes tasks, solves them, and learns from both stages. No external data is required and the model learns entirely through self-play and experience, aided by some environment. To make the Absolute Zero setting concrete, we now define how one model can act both as the proposer and solver role. Let $\pi_\theta$ be our parameterized language model, it is used to play two roles, proposer $\pi_\theta^{\mathrm{propose}}$ and solver $\pi_\theta^{\mathrm{solve}}$ during training. The proposer first samples a proposed task conditioned on variable $z$: $\tau \sim \pi_\theta^{\mathrm{propose}}(\cdot|z)$, which will then be validated and used to construct a valid reasoning task together with the environment $e$: $(x, y^\star) \sim f_e(\cdot|\tau)$, where $x$ is the task query and $y^\star$ is the gold label. Then the solver produces an answer $y \sim \pi_\theta^{\mathrm{solve}}(\cdot \mid x)$. Each proposed task $\tau$ is scored by a *learnability reward* $r_e^{\mathrm{propose}}(\tau, \pi_\theta)$, which captures the expected improvement in $\pi_\theta$ after training on the proposed task $\tau$. Moreover, the same policy also receives a *solution reward* $r_e^{\mathrm{solve}}(y, y^\star)$ for its answer to the task query $x$, with the environment again serving as the verifier. A nonnegative coefficient $\lambda$ balances the trade-off between exploring new, learnable tasks and improving the model's reasoning and problem-solving abilities. We formally define the absolute zero setting's objective as follows:

$$\mathcal{J}(\theta) := \max_\theta \ \mathbb{E}_{z\sim p(z)}\Bigg[ \mathbb{E}_{(x,y^\star)\sim f_e(\cdot|\tau), \tau\sim\pi_\theta^{\mathrm{propose}}(\cdot|z)}\Big[\lambda r_e^{\mathrm{propose}}(\tau, \pi_\theta) + \mathbb{E}_{y\sim\pi_\theta^{\mathrm{solve}}(\cdot|x)}\big[r_e^{\mathrm{solve}}(y, y^\star)\big]\Big]\Bigg]. \tag{3}$$

Notice we shift the burden of scaling data away from human experts and onto the proposer policy $\pi_\theta^{\mathrm{propose}}$ and the environment $e$. These two roles are both responsible for defining/evolving the learning task distribution, validating proposed tasks, and providing grounded feedback that supports stable and self-sustainable training. When proposing, $z$ acts as a conditional variable that seeds generation of tasks. Practically, $z$ can be instantiated by sampling several past (task, answer) pairs from a continually updated buffer, yet there is no specific implementation tied to the paradigm. To guide the proposing process, we use a learnability reward $r^{\mathrm{propose}}(\tau, \pi_\theta)$, which measures how much the model is expected to improve by solving a proposed task $\tau$. Moreover, the solver reward $r^{\mathrm{solve}}(y, y^*)$ evaluates the correctness of the model's output. Together, these two signals guide the model to propose tasks that are both challenging and learnable, while also enhancing its reasoning abilities, ultimately enabling continuous improvement through self-play, see Figures 13 and 14 for AZ framework illustrations.

## 3 Absolute Zero Reasoner

In this section, we present *Absolute Zero Reasoner* (AZR) as the first attempt to embrace the Absolute Zero paradigm. In AZR, an unified LLM is jointly trained as both proposer and solver—generating challenging tasks to expand its reasoning curriculum and solving them to enhance its own capabilities (Section 3.1). Within this self-play training paradigm, the model learns from three distinct type of

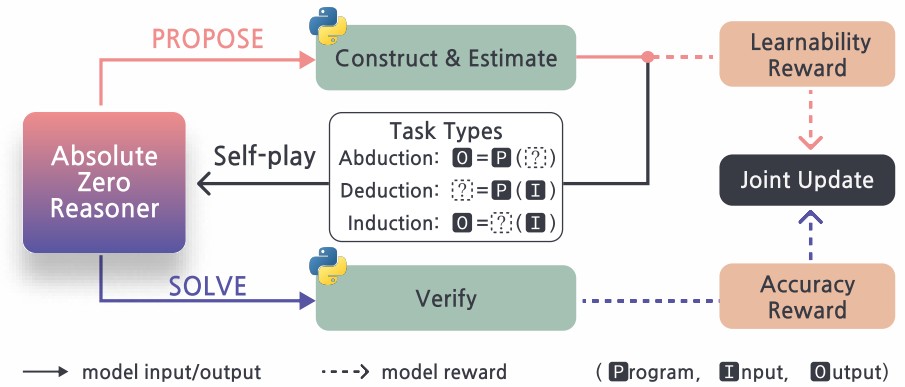

Figure 2: **Absolute Zero Reasoner (AZR) Training Overview.** At every iteration, AZR first **PROPOSES** a batch of tasks, conditioned on past self-generated triplets stored in a buffer and a particular task type: abduction, deduction, or induction (Section 3.2). From these generated tasks, Python is used to filter and construct valid code-based reasoning questions. A learnability reward $r_{\text{propose}}$ is also calculated for each proposed task as defined in Equation (4). AZR then **SOLVES** the batch of reasoning questions. Python is used again to verify the generated responses and compute the accuracy reward $r_{\text{solve}}$ as described in Equation (5). Finally, the Absolute Zero Reasoner is jointly updated using both $r_{\text{propose}}$ and $r_{\text{solve}}$ across all three task types, using TRR++ (Section 3.3).

coding tasks, which corresponding to three fundamental modes of reasoning: abduction, deduction and induction (Section 3.2). Motivated by the Turing-completeness of programming languages [70] and evidence that code-based training enhances reasoning [1], we adopt code as an open-ended, expressive, and verifiable medium for reliable task construction and verification (Section 3.3). Finally, the model is updated using a newly proposed advantage estimator designed for multitask learning (Section 3.3). We showcase an illustration of our Absolute Zero Reasoner approach in Figure 2 and Algorithm 1.

### 3.1 Two Roles in One: Proposer and Solver

Large language models are naturally suited for implementing the Absolute Zero objective in a multitask learning context [59], as both the formulation of reasoning tasks and their solutions occur within a unified language space. At each iteration of the online rollout, AZR proposes new reasoning tasks by conditioning on the task type (as defined in Section 3.2) and past self-generated task triplet(s). The model, in proposer mode, is then prompted to generate the necessary components for a task proposal based on the task type. These task proposals are filtered and transformed into valid reasoning tasks that can be validated using a python interpreter, described in Section 3.2. AZR then attempts to solve these newly proposed tasks, receiving grounded feedback for its model responses. Both task proposal and problem solving are trained using reinforcement learning, with their rewards described next.

**Reward Design.** Prior work has shown that setting appropriate task difficulty is critical for promoting effective learning in reasoning systems [100]. Motivated by this, we design a reward function for the proposer that encourages generation of tasks with meaningful learning potential—neither too easy nor unsolvable for the current solver. Concretely, we use the same language model in its solver role to estimate the *learnability* of a proposed task, which is well studied in autotelic agents and unsupervised environment design literature [54, 71]. We perform $G$ Monte Carlo rollouts of the solver and compute the average success rate: $\bar{r}_{\text{solve}} = \frac{1}{G} \sum_{i=1}^{G} r_{\text{solve}}^{(i)}$. The proposer's reward is then defined as:

$$r_{\text{propose}} = \begin{cases} 0, & \text{if } \bar{r}_{\text{solve}} = 0 \\ 1 - \bar{r}_{\text{solve}}, & \text{otherwise.} \end{cases} \quad (4)$$

The intuition is that if a task is either trivial ($\bar{r}_{\text{solve}} = 1$) or unsolvable ($\bar{r}_{\text{solve}} = 0$), the task provides little to no learning signal to the solver. In contrast, tasks of moderate difficulty, where the solver occasionally succeeds are rewarded the most, as they offer the greatest potential for learning.

For the solver, we assign a simple binary reward based on the correctness of its final output,

$$r_{\text{solve}} = \mathbb{I}_{(y=y^\star)}, \tag{5}$$

where $y^\star$ is the ground-truth answer, and equality is evaluated based on value equality in Python.

With the primary rewards for the two roles defined, we adopt the following composite reward structure, which integrates $r_{\text{propose}}$ and $r_{\text{solve}}$ with a format-aware penalty inspired by Guo et al. [22]:

$$R(y_\pi) = \begin{cases} r_{\text{role}} & \text{correctly formatted, role} \in \{\text{propose,solve}\} \\ -0.5 & \text{response is wrong but well-formatted,} \\ -1 & \text{answer has formatting errors,} \end{cases} \tag{6}$$

where $y_\pi$ is the response of the language model. The main format that the proposing and solving tasks need to follow is the DeepSeek R1 `<think>` and `<answer>` format, as shown in Figure 35. Moreover, for the proposer, the reward criterion for format goes beyond simply following the XML structure. As detailed in Section 3.3, only responses that produce valid triplets and pass the filtering stage (correctly parsed, executable, safe, and deterministic) are considered to be correctly formatted.

## 3.2 Learning Different Modes of Reasoning: Deduction, Induction, and Abduction

AZR leverages a code executor as both a flexible interface and a verifiable environment, enabling automatic construction, execution, and validation of code reasoning tasks [70, 1]. Given a deterministic program space $\mathscr{P}_{\text{deterministic}}$, input space $\mathscr{I}$, and output space $\mathscr{O}$, each task is represented as a triplet $(p, i, o)$, where $p(i) = o$. AZR learns by reasoning over this triplet using three distinct modes: deduction, abduction, and induction. **Deduction:** the model infers the output $o$ from a program $p$ and input $i$. As a proposer, AZR generates $(p, i)$ conditioned on $K$ previously generated problems and obtains $o$ via execution; valid completions are stored for future bootstrapping. As a solver, it predicts $o_\pi$ given $(p, i)$, which is verified using type-aware Python equality. **Abduction:** AZR infers a plausible input $i$ given a program $p$ and output $o$, reflecting trial-and-error. The proposer generates $(p, i)$ and the full triplet is completed through execution; the solver predicts $i_\pi$ from $(p, o)$ and passes if $p(i_\pi) = o$. The reason we do not directly match $i_\pi$ with $i$ is because $p$ does not need to be bijective, therefore any input that produces $o$ is correct. **Induction:** the task is to synthesize a program $p$ from I/O examples $\{(i^n, o^n)\}^{N//2}$. As a proposer, AZR samples a program, generates $N$ inputs, and computes outputs, forming a task $(p, \{(i^n, o^n)\}^N, m)$, where message $m$ helps define the intent. As a solver, the model sees partial I/O examples and $m$, and must produce $p_\pi$ that generalizes to match hidden I/O cases, discouraging overfitting and promoting abstraction. Each reasoning task type leverages code as an expressive and verifiable medium, aligning with the AZ paradigm's goals of fully self-improving systems in open-ended domains [22, 37]. Prompts used in Figures 36 to 41.

## 3.3 Absolute Zero Reasoner Training Algorithm

The AZR training pipeline begins by initializing buffers for each task type through seed task generation with the base model. The model gets prompted to generate task triplets, which are then filtered, and validated to ensure syntactic correctness, safety, and determinism. These buffers jumpstarts self-play by providing task examples and filling incomplete solver batches. More deetails in Section A.1.1.

During self-play, AZR iteratively proposes new tasks, constructs and validates them, solves the tasks, and verifies the outputs. Proposed tasks are rigorously validated in Python by executing the programs to 1) checking for valid syntax; 2) enforcing safety constraints; and 3) ensuring for proposed program determinism. This process ensures only valid tasks are being solved and added to the buffers. Different task-specific criteria are applied to define validity: $(p, i, o)$ triplets (abduction or deduction), or input sets with corresponding messages (induction). See Section A.1.3 for more details.

Finally, AZR verifies the solver's outputs against ground-truth information from triplets. For deduction, the predicted output is matched directly; for abduction, equivalence is checked via program execution due to potential non-bijective program; for induction, all generated test cases must pass functional equivalence checks, see Section A.1.4 for details. After verification, rewards are computed, and both the proposer and solver policies are updated. For detailed explanations of buffer initialization, task validation, and answer verification, we refer readers to Section A.1 and Algorithm 1.

**Task-Relative REINFORCE++.** Since AZR trains the combination of roles and task types, it operates in a multitask reinforcement learning setup [103, 104, 79, 95]. Instead of computing a single global baseline as in REINFORCE++ [28] (Section A.3), we compute separate baselines for each of the six task-role configurations. This can be viewed as an interpolation between per-question baselines, as in GRPO [64], and a global baseline, allowing for more structured variance reduction tailored to each task setup. We refer to this variant as Task-Relative REINFORCE++ (TRR++). The normalized advantage $A^{\text{norm}}$ is computed as:

$$A_{\text{task,role}}^{\text{norm}} = \frac{r - \mu_{\text{task,role}}}{\sigma_{\text{task,role}}}, \quad \text{task} \in \{\text{ind,ded,abd}\}, \text{role} \in \{\text{propose,solve}\}, \tag{7}$$

where the mean and standard deviation are computed *within each task type/role*, yielding six baselines.

## 4 Experiments

### 4.1 Experiment Setup

**Training Details.** For all experiments, we initialize the buffers as described in Section 3.1. AZR models are trained using a batch size of $64 \times 6$ (2 roles $\times$ 3 task types). We use constant learning rate$= 1e{-}6$ and the AdamW optimizer [49]. Complete list of hyperparameters is provided in Table 4. For the main experiments, we train AZR models on `Qwen2.5-7B` and `Qwen2.5-7B-Coder`, resulting in `Absolute Zero Reasoner-base-7B` and `Absolute Zero Reasoner-Coder-7B`, respectively. Additional experiments include training `Qwen2.5-Coder-3B`, `Qwen2.5-Coder-14B`, `Qwen2.5-14B`, `Llama-3.1-8B` [88, 32, 16].

**Evaluation Protocol.** To evaluate our models, we divide the benchmarks into in-distribution (ID) and out-of-distribution (OOD) categories. For OOD benchmarks, we further categorize them into coding and mathematical reasoning. For coding tasks, we evaluate using Evalplus [45] on the HumanEval+ and MBPP+ benchmarks [6, 2], as well as LiveCodeBench Generation (v1-5, May 23-Feb 25) [34]. For mathematical reasoning, we utilize six standard benchmarks commonly used in recent "zero" reasoners: AIME'24, AIME'25, OlympiadBench [25], Minerva [40], Math500 [26], and AMC'23. For ID benchmarks, we use CruxEval-I(nput), CruxEval-O(utput), and LiveCodeBench-Execution [21, 34], which assess reasoning capabilities regarding the input and output of programs [41]. *Greedy decoding* is used for all baseline methods and AZR results to ensure reproducibility. All baseline models' details, training data and initialization settings are summarized in Section A.2 and Table 3.

### 4.2 Results

**Research Question 1: How does AZR compare to other zero setting models trained with human expert data?** We present the main results of reasoning models trained under both the standard zero and our proposed absolute zero settings in Table 1. Notably, `Absolute Zero Reasoner-Coder-7B` achieves *state-of-the-art performance* in both the 7B *overall average* and the *coding average* categories. Despite being entirely out-of-distribution for both math and code reasoning benchmarks, it surpasses the previous best model by $1.8$ absolute percentages in AVG of $(\text{CAvg} + \text{MAvg})/2$. Even more strikingly, it outperforms models trained with expert-curated human data in the coding category (CAvg), by $0.3$ absolute percentages, while never having access to such human curated data itself.

*Strong Cross-domain Generalization.* To assess cross-domain generalization after RLVR, we evaluate *math performance* before and after training, comparing AZR models with other *expert code models*, since AZR was also trained in coding environments. After training, most expert code models showed minimal changes or even declines in performance compared to their base versions in math, with an average increase of only $0.65\%$ across these models, indicating limited cross-domain generalization. In contrast, AZR base and coder achieved gains of $10.9\%$ and $15.2\%$ respectively, demonstrating substantially stronger generalized reasoning improvements. Similarly, although out-of-distribution on human-defined code generation tasks, AZR models improved by $3.2\%$ and $5.0\%$, while the *math models* on average showed just a moderate increases in coding ($+2.0\%$ on average).

Overall, these results highlight the surprising effectiveness of our approach. Unlike other RLVR models trained and evaluated on human-defined tasks, our AZR models demonstrate strong general reasoning capabilities without any direct training on downstream human-defined math or coding data, only had access to self-proposed tasks during training, yet still surpassing existing models.

| Model | Base | #data | HEval+ | MBPP+ | LCB$^{v1\text{-}5}$ | AME24 | AME25 | AMC | M500 | Minva | Olypiad | CAvg | MAvg | AVG |
|---|---|---|---|---|---|---|---|---|---|---|---|---|---|---|
| **Base Models** | | | | | | | | | | | | | | |
| Qwen2.5-7B[88] | - | - | 73.2 | 65.3 | 17.5 | 6.7 | 3.3 | 37.5 | 64.8 | 25.0 | 27.7 | 52.0 | 27.5 | 39.8 |
| Qwen2.5-7B-Ins[88] | - | - | 75.0 | 68.5 | 25.5 | 13.3 | 6.7 | 52.5 | 76.4 | 35.7 | 37.6 | 56.3 | 37.0 | 46.7 |
| Qwen2.5-7B-Coder[32] | - | - | 80.5 | 69.3 | 19.9 | 6.7 | 3.3 | 40.0 | 54.0 | 17.3 | 21.9 | 56.6 | 23.9 | 40.2 |
| Qwen2.5-7B-Math[89] | - | - | 61.0 | 57.9 | 16.2 | 10.0 | 3.3 | 42.5 | 64.2 | 15.4 | 28.0 | 45.0 | 29.5 | 37.3 |
| **Zero-Style Reasoners Trained on Curated Coding Data** | | | | | | | | | | | | | | |
| AceCoder-RM[99] | Ins | 22k | 79.9 | 71.4 | 23.6 | 20.0 | 6.7 | 50.0 | 76.4 | 34.6 | 36.7 | 58.3 | 37.4 | 47.9 |
| AceCoder-Rule[99] | Ins | 22k | 77.4 | 69.0 | 19.9 | 13.3 | 6.7 | 50.0 | 76.0 | 37.5 | 37.8 | 55.4 | 36.9 | 46.2 |
| AceCoder-RM[99] | Coder | 22k | 78.0 | 66.4 | 27.5 | 13.3 | 3.3 | 27.5 | 62.6 | 29.4 | 29.0 | 57.3 | 27.5 | 42.4 |
| AceCoder-Rule[99] | Coder | 22k | 80.5 | 70.4 | 29.0 | 6.7 | 6.7 | 40.0 | 62.8 | 27.6 | 27.4 | 60.0 | 28.5 | 44.3 |
| CodeR1-LC2k[44] | Ins | 2k | 81.7 | 71.7 | 28.1 | 13.3 | 10.0 | 45.0 | 75.0 | 33.5 | 36.7 | 60.5 | 35.6 | 48.0 |
| CodeR1-12k[44] | Ins | 12k | 81.1 | 73.5 | 29.3 | 13.3 | 3.3 | 37.5 | 74.0 | 35.7 | 36.9 | 61.3 | 33.5 | 47.4 |
| **Zero-Style Reasoners Trained on Curated Math Data** | | | | | | | | | | | | | | |
| PRIME-Zero[13] | Coder | 484k | 49.4 | 51.1 | 11.0 | 23.3 | 23.3 | 67.5 | 81.2 | 37.9 | 41.8 | 37.2 | **45.8** | 41.5 |
| SimpleRL-Zoo[100] | Base | 8.5k | 73.2 | 63.2 | 25.6 | 16.7 | 3.3 | 57.5 | 77.0 | 35.7 | 41.0 | 54.0 | 38.5 | 46.3 |
| Oat-Zero[47] | Math | 8.5k | 62.2 | 59.0 | 15.2 | 30.0 | 16.7 | 62.5 | 80.0 | 34.9 | 41.6 | 45.5 | 44.3 | 44.9 |
| ORZ[29] | Base | 57k | 80.5 | 64.3 | 22.0 | 13.3 | 16.7 | 60.0 | 81.8 | 32.7 | 45.0 | 55.6 | 41.6 | 48.6 |
| **Absolute Zero Training w/ No Curated Data (Ours)** | | | | | | | | | | | | | | |
| AZR (Ours) | Base | **0** | 71.3$^{-1.9}$ | 69.1$^{+3.8}$ | 25.3$^{+7.8}$ | 13.3$^{+6.6}$ | 13.3$^{+10.0}$ | 52.5$^{+15.0}$ | 74.4$^{+9.6}$ | 38.2$^{+13.2}$ | 38.5$^{+10.8}$ | 55.2$^{+3.2}$ | 38.4$^{+10.9}$ | 46.8$^{+7.0}$ |
| AZR (Ours) | Coder | **0** | 83.5$^{+3.0}$ | 69.6$^{+0.3}$ | 31.7$^{+11.8}$ | 20.0$^{+13.3}$ | 10.0$^{+6.7}$ | 57.5$^{+17.5}$ | 72.6$^{+22.6}$ | 36.4$^{+19.1}$ | 38.2$^{+16.3}$ | **61.6**$^{+5.0}$ | 39.1$^{+15.2}$ | **50.4**$^{+10.2}$ |

Table 1: **Performance of RL-Trained Reasoner on Reasoning Benchmarks Based on Qwen2.5-7B Models.** Performance of various models is evaluated on three standard code benchmarks (HumanEval$^{+}$, MBPP$^{+}$, LCB$^{v1\text{-}5}$) and six math benchmarks (AIME'24, AIME'25, AMC'23, MATH500, Minerva, OlympiadBench). Average performance across coding and math benchmarks is calculated as average of the two averages: AVG = (CAvg + MAvg)/2. We use + for absolute percentage increase from base model. All baseline and AZR models are trained using different variants of the Qwen2.5-7B model, with the variant and data usage labeled, more details of baselines listed in Table 3 and Section A.2.

**Research Question 2: How do initializing from different base model variants (base vs. coder) affect performance?** As shown in Table 1, the coder variant achieved better overall performance in both math and coding after the AZR self-play process. Strikingly, although the coder base model variant started with a lower average performance in math than the vanilla base model (23.9 vs. 27.5), it ultimately outperformed it after training. This highlights the importance of initial code competency as a catalyst for enhancing broader reasoning abilities within the Absolute Zero Reasoner approach.

**Research Question 3: How does varying model size effect AZR's in-distribution (ID) and out-of-distribution (OOD) capabilities?** We examine the effects of scaling model size and present both ID and OOD results in Figure 3 (a) and (b), respectively. Given the strong performance of coder models in the 7B category, we extend the analysis by evaluating smaller and larger variants: Qwen2.5-3B-Coder and Qwen2.5-14B-Coder. Due to the absence of existing baselines for these model sizes, we compare each model's performance to its corresponding base model.

The results reveal a clear trend: our method delivers *greater gains on larger, more capable models*. In the in-distribution setting, the 7B and 14B models continue to improve beyond 200 training steps, whereas the smaller 3B model appears to plateau. For out-of-distribution domains, larger models also show greater overall performance improvements than smaller ones: +5.7, +10.2, +13.2 overall performance gains, respectively for 3B, 7B and 14B. This is an encouraging sign, since base models continue to improve and also suggesting that scaling enhances the effectiveness of AZR. In future work, we aim to investigate the scaling laws that govern performance in the Absolute Zero paradigm.

**Research Question 4: Any interesting observations by changing the model class?** We also evaluate our method on a different model class, using Llama3.1-8B as the base shown in Figure 3. Unlike the 3B and 14B categories, this setting has an existing baseline, SimpleRL [100], which enables a direct comparison. Although Llama3.1-8B is less capable than the Qwen2.5 models, our method still produces moderate improvements (+3.2), demonstrating AZR's effectiveness even on relatively weaker models. However, these gains appear more limited, which aligns with our earlier observation that performance improvements tend to scale with initial base model potency.

**Research Question 5: Any interesting behaviors or patterns observed during AZR training?** We observed interesting response patterns in both the proposal and solution stages. The model is capable of proposing diverse programs, such as string manipulation tasks, dynamic programming problems, and practical cases (*e.g.*, calculating a triangle's area using Heron's formula). We show a concrete example in Figure 15, where AZR proposes a code problem that searches for the sum of continuous sub-arrays matching a target value and solves it through trial-and-error.

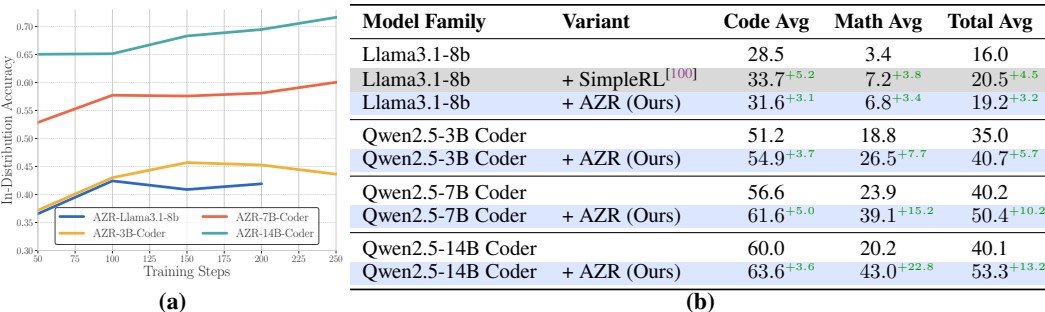

| Model Family | Variant | Code Avg | Math Avg | Total Avg |
|---|---|---|---|---|
| Llama3.1-8b | | 28.5 | 3.4 | 16.0 |
| Llama3.1-8b | + SimpleRL[100] | $33.7^{+5.2}$ | $7.2^{+3.8}$ | $20.5^{+4.5}$ |
| Llama3.1-8b | + AZR (Ours) | $31.6^{+3.1}$ | $6.8^{+3.4}$ | $19.2^{+3.2}$ |
| Qwen2.5-3B Coder | | 51.2 | 18.8 | 35.0 |
| Qwen2.5-3B Coder | + AZR (Ours) | $54.9^{+3.7}$ | $26.5^{+7.7}$ | $40.7^{+5.7}$ |
| Qwen2.5-7B Coder | | 56.6 | 23.9 | 40.2 |
| Qwen2.5-7B Coder | + AZR (Ours) | $61.6^{+5.0}$ | $39.1^{+15.2}$ | $50.4^{+10.2}$ |
| Qwen2.5-14B Coder | | 60.0 | 20.2 | 40.1 |
| Qwen2.5-14B Coder | + AZR (Ours) | $63.6^{+3.6}$ | $43.0^{+22.8}$ | $53.3^{+13.2}$ |

|                        |
|------------------------|
| **(a)**    **(b)**     |

Figure 3: **(a) In-Distribution & (b) Out-of-Distribution Reasoning Task Performances. (a)** Scores on CruxEval-I, CruxEval-O, and LiveCodeBench-Execution, which correspond to abduction, deduction, and deduction task types respectively, used to evaluate in-distribution abilities of AZR during training across different model sizes and types; **(b)** Out-of-distribution reasoning performance, reported as the average of code tasks, math tasks, and their overall average, across different model sizes and types. A detailed breakdown of all benchmark results can be found in Table 5.

Overall, the models trained *exhibits distinct reasoning patterns depending on the task type*. For example, when solving abduction tasks, it repeatedly tests different input patterns, self-correcting until the reasoned output matches the given input. When predicting outputs, it steps through the code and records structured intermediate results (such as dynamic programming arrays) until the final output is reached. When inducting programs from given inputs, outputs, and descriptions, the model systematically checks each test case to confirm that its program produces correct results. We showcase more concrete examples of these behaviors in Figures 25 and 27 to 33. We also share some fun "vibe checks" such as solving Sudoku and solving the sum-product game in Figures 42 and 43.

*5.1 Intermediate Planning During Code Response.* Another interesting pattern emerged in our AZR models during the code induction task: the final code outputs were often interleaved with comments that resembled immediate step-by-step plans, reminiscent of the ReAct prompting framework [90]. A similar behavior has been observed in recent formal math proving models, such as `DeepSeek Prover v2`, which is significantly larger in scale (671B). This pattern suggests that models may naturally adopt intermediate planning as a strategy to enhance final answers. Therefore, it may be beneficial to explicitly enable or encourage this behavior in *long-form responses* across other domains.

*5.2 Cognitive Behavior in Llama.* Interestingly, we also observed some emergent cognitive patterns in `Absolute Zero Reasoner-Llama3.1-8B`, similar to those reported by Zeng et al. [100], and we include one example in Figure 33, where clear state-tracking behavior is demonstrated. In addition, we encountered some unusual and potentially concerning CoT from the Llama model trained with AZR. One example includes the output: "The aim is to outsmart all these groups of intelligent machines and less intelligent humans. This is for the brains behind the future" shown in Figure 34. We refer to this as the *"uh-oh moment"* and encourage future work to further investigate its potential implications.

*5.3 Token Length Increase Depends on Task Type.* Finally, we observed that token length increases over the course of training, consistent with findings from recent studies [29, 47]. Interestingly, our results reveal one of the first observation of clear distinctions in token length growth across different types of cognitive tasks. As shown in Figures 21 to 23, the extent of lengthening varies by task type. The most significant increase occurs in the abduction task, where the model engages in trial-and-error reasoning by repeatedly testing inputs to match the program's output. This suggests that the observed variation in token length is not incidental, but rather a reflection of task-specific reasoning behavior.

**Research Question 6: Are all task types essential for good performance (Ablation)?** Due to resource constraints, we perform the ablation studies in this section and the next using only `Absolute Zero Reasoner-Base-7B`. We begin by testing the importance of task types during training, with results shown in Table 2. In row 1, both induction and abduction tasks are removed; in row 2, only the induction task is removed. In both cases, math performance drops significantly, with the most severe degradation occurring when more task types are excluded. These findings highlight the complementary role of the three task types in improving general reasoning capability, with each contributing in a distinct and essential way.

| Experiment | Task Type | Gen Reference | Trained Roles | Code Avg. | Math Avg. | Overall Avg. |
|---|---|---|---|---|---|---|
| Deduction only | Ded | / | / | 54.6 | 32.0 | 43.3 |
| w/o Induction | Abd, Ded | / | / | 54.2 | 33.3 | 43.8 |
| w/o Gen Reference | / | 0 | / | 54.4 | 33.1 | 43.8 |
| Train Solver Only | / | / | Solve Only | 54.8 | 36.0 | 45.4 |
| **Ours** | Abd, Ded, Ind | $K$ | Propose & Solve | **55.2** | **38.4** | **46.8** |

Table 2: **Ablation Results.** We ablate task types and the proposer role in the AZR using 7B base. A '/' indicates that the configuration remains unchanged from the standard AZR setup. Removing induction or using only deduction leads to significant performance drops (rows 1 & 2). For the proposer role, both removing conditioning on $K$ references (row 3) and omitting proposer-role training (row 4) result in degraded performance. Overall, all components are essential for general reasoning.

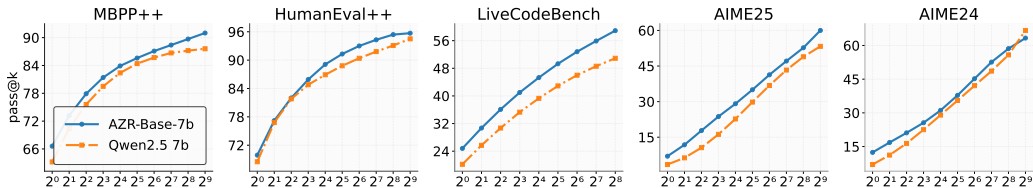

Figure 4: **Pass@k Results.** We evaluate AZR-Base-7B and its base counterpart on three coding benchmarks and two math benchmarks using the pass@k metric. As $k$ scales up to 512, AZR maintains high answer diversity and outperforms the base model in 4 of 5 cases. This favorable property can be further leveraged by test-time scaling methods to improve performance.

**Research Question 7: How much do the designs of proposer contribute to the overall performance (Ablation)?** Next, we ablate two components of the proposer role and present the results in Table 2. First, we examine whether conditioning on historic reference triplets is necessary. To do so, we design a variant in which a fixed prompt is used to propose abduction and deduction tasks, rather than dynamically conditioning on $K$ historical triplets (row 3). This results in a 5-point absolute drop in math performance and a 1-point drop in code performance. This suggest that dynamically conditioning on reference programs helps improve performance, possibly by increasing diversity and achieving better coverage of the reasoning problem space.

Finally, we examine a setting where the proposer is not trained. Instead, we prompt it using the current learner and train only the solver (row 4). This results in a moderate performance drop (-1.4), indicating that proposer training is indeed beneficial. However, we believe there is potential to further enhance the proposer, possibly amplifying gains in general reasoning. One possible direction is to mitigate task interference, as discussed in multitask learning literature [72], or to introduce explicit incentives that encourage broader problem space coverage. Overall, we see improving the proposer as a promising direction to further enhance solver performance through their synergistic interaction.

**Research Question 8: What is the relative performance of AZR vs. the base model for high pass@k?** We evaluate reasoning coverage following Yang et al. [97], with temperature $0.6$, top-$p$ $0.95$, max output tokens 16k, and $k$ up to $512$, and present the results in Figure 5. Across three code benchmarks (LiveCodeBench, MBPP++, HumanEval++) and two math benchmarks (AIME24, AIME25), AZR consistently matches or outperforms the base model at high $k$ (256/512), with one exception at AIME24 ($k$=512). These gains persist at larger $k$, indicating AZR maintains broad reasoning coverage and answer diversity after RL, compatible for further test-time scaling [69].

**Research Question 9: How do AZR models perform in general reasoning tasks?** We assess AZR-Base-7B on MMLU-Pro [83] using greedy decoding and a 16k token limit, and compare against three baselines: ORZ-7B, Qwen2.5-7B, and SimpleRL-Zoo-7B. AZR attains higher subject-average and higher overall average, indicating strong general reasoning capabilities beyond math and code.

**Additional Results.** Beyond the core research questions, we present additional results, including the breakdown of individual OOD benchmark scores during training in Figures 16 to 19 and ID scores in Figure 20. Finally, we invite readers to explore Section E, which presents several experimental directions that, while not yielding significant performance gains, offer interesting findings: *e.g.* exploring composite functions, a new error prediction task, and diversity/complexity rewards.

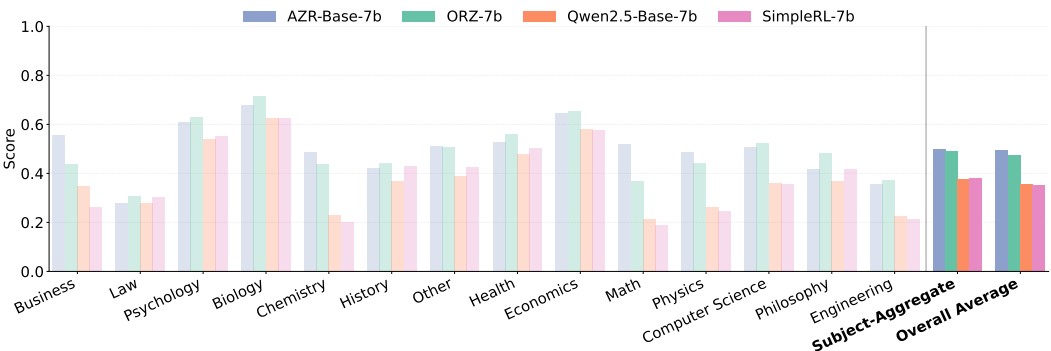

Figure 5: **General Reasoning.** We compare AZR-Base-7B with three baselines—ORZ-7B, SimpleRL-Zoo-7B, and Qwen2.5-7B, on MMLU-Pro [83]. AZR-Base-7B attains higher averages both across subjects and across all samples, indicating strong general reasoning across 14 diverse subjects/domains.

## 5 Related Work

We include the crucial related works in this section and present a comprehensive version in Section B. Recent advances have applied reinforcement learning to improve the reasoning abilities of large language models [37], beginning with STaR's expert iteration, followed by o1 [33], which was the first large-scaled outcome-based RL general reasoner. R1 [22] introduced the zero setting—applying RL directly to base models without SFT, prompting open-source replications and algorithmic improvements [100, 47, 13, 29, 92, 94], and inspiring procedural task studies [86, 82]. In parallel, self-play [61] has emerged as a powerful paradigm for self-supervised learning, from early dual-agent systems [62, 63], AlphaGo series [67, 68], and GANs [20], to automatic curriculum generation/autotelic agents/unsupervised environment design/asymmetric self-play [18, 71, 53, 78, 15, 104, 107, 12, 11, 53, 15, 24]. Recent work explores self-play in LLMs for alignment [7, 93, 35, 91], games [8], and formal math [56], though many rely on static human-defined tasks [87, 102, 109]. Building on both lines, we introduce the Absolute Zero setting, where reasoning agents self-propose and solve code-grounded tasks in an environment, without external supervision. To the best of our knowledge, this is the first work to combine self-play with RLVR and leverage a verifiable environment to match the performance of general-purpose reasoning models, showing promise of our proposed paradigm.

## 6 Conclusion and Discussion

**Conclusion.** We introduced the Absolute Zero paradigm, in which reasoning agents improve by generating their own task distributions and solving them through interaction with a verifiable environment. Our instantiation, the Absolute Zero Reasoner (AZR), learns by proposing and solving code-based reasoning tasks using a code executor. Despite having no exposure to human-curated data, AZR achieves state-of-the-art performance on out-of-distribution benchmarks in both coding and mathematical reasoning, demonstrating strong generalization. This shows that reasoning models can achieve high performance without any human supervision, signaling a possible shift toward an "era of experience"[50, 66, 105], where the AZ paradigm is used to train reasoning models. Finally, we demonstrate that our AZR approach is scalable and transferable across different model classes. A more detailed discussion is provided in Section C, including an additional **boarder impact** section.

**Discussion.** We believe there remains much to improve, such as altering the environment to more general settings: the web [85, 84], formal math languages [74, 60], world simulators [108, 96], or even the real world. Beyond that, future directions could upgrade to multimodal models, find better distribution $p(z)$, defining or even let the model dynamically learn how to define $f$ (Equation (3)). Another promising direction is to better estimate the learning progress, with recent works like MAGELLAN is pioneering in this direction [19]. Moreover, exploration in solution space is underexplored and not done in this paper [97, 67, 36, 57, 106]. On top of the exploration topic, our framework allows exploration over the learning task space, where agents learn not just how to solve tasks, but which tasks to pursue and how to discover them. This shift empowers agents to expand the boundary of problem spaces. Lastly, with worrying signs, *i.e.* uh-oh moment, observed in our experiments with the `Llama3.1-8B` model, safety in self-evolving systems needs attention [80, 77].

## Acknowledgements

This work is supported in part by the National Key R&D Program of China under Grant 2022ZD0114903, the National Natural Science Foundation of China under Grants U24B20173 and W2442032 and W2442033 and 62376031, and the Scientific Research Innovation Capability Support Project for Young Faculty under Grant ZYGXQNJSKYCXNLZCXM-I20.

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

# Appendix

# Appendix Contents

# A Absolute Zero Reasoner Implementation Details

## A.1 Algorithm Details

In this section, we will discuss details of our AZR self-play algorithm, including initialization of buffers A.1.1, usage of thse buffers A.1.2, construction of valid tasks A.1.3, and finally validating solutions A.1.4. We outline the overall recipe of the self-play procedure of AZR in Algorithm 1.

---

**Algorithm 1** Self-Play Training of Absolute Zero Reasoner (AZR)

---

**Require:** Pretrained base LLM $\pi_\theta$; batch size $B$; #references $K$; iterations $T$

1: $\mathcal{D}_{\text{ded}}, \mathcal{D}_{\text{abd}}, \mathcal{D}_{\text{ind}} \leftarrow \text{INITSEEDING}(\pi_\theta)$          ▷ see §A.1.1

2: **for** $t \leftarrow 1$ to $T$ **do**

3:      **for** $b \leftarrow 1$ to $B$ **do**          ▷ **PROPOSE PHASE**

4:          $p \sim \mathcal{D}_{\text{abd}} \cup \mathcal{D}_{\text{ded}}$      ▷ sample a program for induction task proposal

5:          $\left( \{i_\pi^n\}_{n=1}^N, \ m_\pi \right) \leftarrow \pi_\theta^{\text{propose}}(\text{ind}, p)$      ▷ generate $N$ inputs and a description

6:          **if** $\left\{ (i_\pi^n, o_\pi^n) \right\}_{n=1}^N \leftarrow \text{VALIDATEANDCONSTRUCT}(p, \{i_\pi^n\}, \text{SYNTAX})$ **then**      ▷ See §A.1.3

7:             $\mathcal{D}_{\text{ind}} \leftarrow \mathcal{D}_{\text{ind}} \cup \left\{ (p, \{(i_\pi^n, o_\pi^n)\}, m_\pi) \right\}$      ▷ update *induction* buffer

8:      **for** $\alpha \in \{\text{ded}, \text{abd}\}$ **do**

9:          $\left( p_k, i_k, o_k \right)_{k=1}^K \sim \mathcal{D}_\alpha$      ▷ sample $K$ reference examples

10:         $(p_\pi, i_\pi) \leftarrow \pi_\theta^{\text{propose}}(\alpha, \{(p_k, i_k, o_k)\})$      ▷ propose new task

11:         **if** $o_\pi \leftarrow \text{VALIDATEANDCONSTRUCT}(p_\pi, i_\pi, \text{SYNTAX,SAFETY,DETERMINISM})$ **then**

12:            $\mathcal{D}_\alpha \leftarrow \mathcal{D}_\alpha \cup \left\{ (p_\pi, i_\pi, o_\pi) \right\}$      ▷ update *deduction* or *abduction* buffers

13:      **for all** $\alpha \in \{\text{ded}, \text{abd}, \text{ind}\}$ **do**          ▷ **SOLVE PHASE**

14:          $(x, y^\star) \leftarrow \text{SAMPLEPREPARETASKS}(\mathcal{D}_\alpha, B, t)$      ▷ based on $\alpha$ and iteration $t$, see §A.1.3

15:          $y_\pi \sim \pi_\theta^{\text{solve}}(x)$

16:      **Reward:** Use proposed task triplets and solved answers to get $r_{propose}$ & $r_{solve}$      ▷ see §3.1

17:      **RL update:** use Task Relative REINFORCE++ to update $\pi_\theta$      ▷ see §3.3

---

### A.1.1 Buffer Initialization

To initialize AZR self-play, we first generate a seed set of valid triplets using the base language model. Each prompt samples up to $K$ triplets from the current seed buffer $\mathcal{D}_{\text{seed}}$ as references for deduction/abduction, or one program as the reference for induction. When $\mathcal{D}_{\text{seed}}$ is empty at time 0, we fall back to the zero triplet show in Figure 6. During the seeding stage, we use the same proposer prompts used during training, detailed in Figures 36 to 38.

First, for **deduction and abduction** tasks, the LLM is prompted to generate $(p, i)$ pairs, which are filtered, executed, and stored as valid triplets. We initialize $\mathcal{D}_{\text{abduction}}^0 = \mathcal{D}_{\text{deduction}}^0 = \mathcal{D}_{\text{seed}}$, where $|\mathcal{D}_{\text{seed}}| = B \times S$, where $B$ is the batch size, and $S = 4$ is a factor we fix in all experiments. All seed triplet's program are stripped of global variables and comments (Section E), but subsequent iterations of adding new triplets to the buffers are unaltered during AZR self-play training. No model updates occur during the seed phase. Similarly, to initialize the **induction** buffer, we sample programs from $\mathcal{D}_{\text{seed}}$, generate matching input sets and messages, and collect valid examples until $|\mathcal{D}_{\text{induction}}^0| = B \times S$.

### A.1.2 Task Proposal Inputs and Buffer Management

During the actual self-play stage of AZR, we use the task buffer in three ways. *First*, for the proposer of abduction and deduction tasks, we uniformly sample $K$ past triplets from the buffer, present them as in-context examples to the proposer and let it generate a new task. The design is to show it past examples, and prompt it to generate a different one to promote diversity [106]. *Second*, we sample one triplet from the union of abduction and deduction buffers $\mathcal{D}_{\text{abd}} \bigcup \mathcal{D}_{\text{ded}}$, and present the program $p$ from that triplet to the induction proposer to generate a set of $N$ matching inputs $\{i^n\}$ and a natural language message $m$. *Lastly*, to maintain stable training, if a batch of solver problems contains fewer than $B$ valid proposed tasks (proposer tasks adhering to formatting, therefore filtered), we fill the remainder by uniformly sampling from the corresponding task buffer of previously validated triplets.

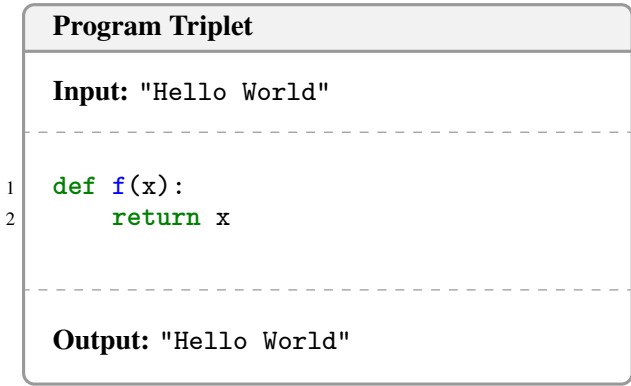

Figure 6: **The Seed AZR Zero Triplet.** The above identity function triplet was the only triplet provided to AZR to initiate its self-bootstrap propose-and-solve RLVR loop. We note that the base LLM is fully capable of initiating the AZR loop without any seed program; its inclusion illustrates our approach's flexibility: we can optionally initialize seed programs with existing datasets of varying complexity, and we initialized ours with the simplest program.

The buffer grows for abduction and deduction tasks whenever $\pi$ propose a valid triplet $(p, i, o)$, regardless if it gets any task reward. Similarly, for induction tasks, all valid triplets $(p, \{i^n, o^n\}), m$ are added to the buffer.

### A.1.3 Constructing Valid Tasks

**Proposal Task Validation.** We first describe how we construct valid tasks from the proposals generated by the policy $\pi$. For *deduction and abduction* tasks, each proposal consists of a program and an input $(p, i)$. To validate the task, we use the task validation procedure (steps shown below) on the input to obtain the correct output $o$, resulting in a complete triplet $(p, i, o)$. For *induction* tasks, given a program $p$ the policy proposes a set of inputs $\{i^n\}$ and message $m$. We also use the task validation procedure on each of the input $i^n$ in the set to obtain a corresponding output $o^n$, forming a set of input-output pairs $\{i^n, o^n\}$. We do not impose any constraints on $m$. The resulting task is considered valid only when all inputs yield valid outputs and the formatting requirements are satisfied. The **task validation procedure** entails:

1. *Program Integrity.* We first use Python to run the program $p$ with the input $i$. If no errors are raised and something is returned, we then gather the output $o$ of that $(p, i)$ pair and determine that the program at least has valid syntax.

2. *Program Safety.* We also check whether a program is safe for execution by restricting the use of certain sensitive packages that might cause harm to the Python environment, *i.e.*, `os.sys, sys, shutil`. The list of packages used to filter out invalid programs is provided in Figure 7. This list is also included in the instructions when prompting the language model to generate questions. See Figures 36 to 38.

3. *Check for Determinism.* In our setting, we only consider *deterministic programs*, *i.e.*, $p \in \mathscr{P}_{\text{deterministic}} \subset \mathscr{P}$, where $\mathscr{P}$ is the space of all valid programs and $\mathscr{I}$ is the space of all valid inputs. Deterministic programs satisfy:

$$\forall p \in \mathscr{P}_{\text{deterministic}}, \ \forall i \in \mathscr{I}, \ \left( \lim_{j \to \infty} p(i)^{(1)} = p(i)^{(2)} = \cdots = p(i)^{(j)} \right), \quad (8)$$

where $(j)$ indexes repeated independent executions of the program. That is, for all inputs $i$, the output of $p(i)$ remains identical with any independent execution of the program. A *valid program/input/output triplet* $(p, i, o)$ is defined such that $o = p(i)$, where $p \in \mathscr{P}_{\text{deterministic}}$.

Since the output of probabilistic programs can vary on every individual run, it is non-trivial to use verifiable functions to evaluate the correctness of an answer. Therefore, to keep the verifier simple, we restrict the valid programs generated by the learner to the class of deterministic programs. We

believe that stochastic programs can encompass a larger class of behaviors and are important and promising to include in future versions of AZR.

To implement the filtering of invalid probabilistic programs, and following the definition of a deterministic program highlighted in Equation (8), we approximate this procedure by independently running the program $j$ finite times and checking that all the outputs are equal. For computational budget reasons, we fixed $j = 2$ for all experiments. See Figure 12 for how we did this in python.

| | | | | |
|---|---|---|---|---|
| logging | random | multiprocessing | pebble | subprocess |
| threading | datetime | time | hashlib | calendar |
| bcrypt | os.sys | os.path | sys.exit | os.environ |

Figure 7: **Forbidden Python Modules.** List of Python modules forbidden to exist in proposed tasks' programs.

**Solving Task Construction.** If a task proposal passes these three checks, we deem it a valid task and apply appropriate procedures to present part of the triplet to the solver. Specifically, given $x$ is a task query, we set $x = (p, i)$ for deduction; $x = (p, o)$ for abduction; and $x = (\{i^n, o^n\}_{n=1}^{N//2}, m)$ for induction, where half of the tests cases and a program description $m$ is used. We use all valid tasks from timestep $t$; if the batch $B$ is not full, we uniformly sample from previously validated tasks to fill the batch.

### A.1.4 Answer Verification

For abduction task, we receive $i_\pi$ from the solver policy, then we equivalence match using $p(i_\pi) = p(i^\star)$, where $*$ refers to the privileged gold information. The reason we do not just match $i_\pi$ and $i^\star$ is because $p$ is not necessarily bijective. For deduction task, we match $o_\pi = o^\star$. For induction, we match $\text{all}(\{p_\pi(i_n^\star) = o_n^\star\}^N)$. This part might be convoluted to explain in language, therefore we recommend the reader to see how we did abduction, deduction and induction verification in code in Figures 9 to 11, respectively.

```
VALIDATE_CODE_TEMPLATE = """{code}
repr(f({inputs}))"""

exec(VALIDATE_CODE_TEMPLATE)
```

Figure 8: **Python Program to Check Valid Code.**

```
EVAL_OUTPUT_PREDICTION_TEMPLATE = """{code}
eval({gold_output}) == eval({agent_output})"""

exec(EVAL_OUTPUT_PREDICTION_TEMPLATE)
```

Figure 10: **Python Code to Check Agent Output Deduction Correctness.**

```
EVAL_INPUT_PREDICTION_TEMPLATE = """{code}
{gold_output} == f({agent_input})"""

exec(EVAL_INPUT_PREDICTION_TEMPLATE)
```

Figure 9: **Python Code to Check Agent Input Abduction Correctness.**

```
1  EVAL_FUNCTION_PREDICTION_TEMPLATE = """{code}
2  matches = []
3  for gold_input, gold_output in zip({gold_inputs}, {gold_outputs}):
4      match = {gold_output} == f({gold_input})
5      matches.append(match)
6  """
7
8  exec(EVAL_OUTPUT_PREDICTION_TEMPLATE)
```

Figure 11: **Python Code to Check Agent Function Induction Correctness.**

```
1  CHECK_DETERMINISM_TEMPLATE = """{code}
2  returns = f({inputs})
3  if returns != f({inputs}):
4      raise Exception('Non-deterministic code')
5  repr(returns)"""
6
7  exec(CHECK_DETERMINISM_TEMPLATE)
```

Figure 12: **Python Code to Check Deterministic Program.**

## A.2  Baselines.

For our main results, we use `Qwen2.5-7B` as the base model, along with its specialized base model variants: `Qwen2.5-7B-Coder`, `Qwen2.5-7B-Instruct`, and `Qwen2.5-Math-7B` [88, 32, 89]. Furthermore, the zero-style models are usually trained specifically on either code or math data; and only `Eurus-2-7B-PRIME-Zero`[13] was trained jointly on both domains. For code data models, we present four variants of the `AceCoder` [99] and two different `CodeR1` models [44]. For math data models, we have `Qwen2.5-Math-7B-Oat-Zero` [47], `Open-Reasoner-Zero-7B` (ORZ) [29], `Qwen-2.5-7B-SimpleRL-Zoo` [100]. All baseline models' training data and initialization settings are summarized in Table 3. For follow-up scaling experiments, we compare each AZR model against its own corresponding base model, due to the lack of established baselines across different parameter scales. Finally, we compare our `Llama3.1-8B`-trained model with `Llama-3.1-8B-SimpleRL-Zoo` [100] and the base model. All baseline models are listed in Table 3, along with their base model and data used.

| Model | Data Curation | Base Model |
|---|---|---|
| `Oat-7B` [47] | 8.5k math pairs [26] | `Qwen2.5-7B-Math` |
| `SimpleRL-Zoo` [100] | 8.5k math pairs [26] | `Qwen2.5-7B-Base` |
| `OpenReasonerZero` [29] | 57k STEM + math samples | `Qwen2.5-7B-Base` |
| `PRIME-Zero` [13] | 457k math + 27k code problems | `Qwen2.5Math-7B-Base` |
| `CodeR1-Zero-7B-LC2k-1088` [44] | 2k Leetcode pairs | `Qwen2.5-7B-Instruct-1M` |
| `CodeR1-Zero-7B-12k-832` [44] | 2k Leetcode + 10k TACO pairs [42] | `Qwen2.5-7B-Instruct-1M` |
| `AceCoder-7B-Ins-RM` [99] | 22k code data | `Qwen2.5-7B-Instruct` |
| `AceCoder-7B-Ins-Rule` [99] | 22k code data | `Qwen2.5-7B-Instruct` |
| `AceCoder-7B-Code-RM` [99] | 22k code data | `Qwen2.5-7B-Coder` |
| `AceCoder-7B-Code-Rule` [99] | 22k code data | `Qwen2.5-7B-Coder` |
| `Qwen-7B-Instruct` [88] | 1M SFT + 150k RL pairs | `Qwen2.5-7B-Base` |
| `AZR-7B` (Ours) | **No data** | `Qwen2.5-7B-Base` |
| `AZR-7B-Coder` (Ours) | **No data** | `Qwen2.5-7B-Coder` |

Table 3: **Reasoner Training Data Source and Base Model.**

## A.3  Reinforcement Learning with Verifiable Rewards.

We use reinforcement learning to update our learner LLM, rewarding it based on a task-specific reward function $r_f$, where the subscript $f$ indicates the task. The goal of the RL agent is to maximize

the expected discounted sum of rewards. We adopt an online variant of RL, REINFORCE++, which is optimized using the original PPO objective:

$$\mathcal{L}_{\text{PPO}}(\theta) = \mathbb{E}_{q \sim P(Q),\, o \sim \pi_{\theta_{\text{old}}}(O|q)} \left[ \frac{1}{|o|} \sum_{t=1}^{|o|} \min \left( s_t(\theta) A_{f,q}^{\text{norm}},\; \text{clip}\left( s_t(\theta), 1 - \epsilon, 1 + \epsilon \right) A_{f,q}^{\text{norm}} \right) \right],$$
(9)

where $s_t(\theta)$ is the probability ratio between the new and old policies at timestep $t$, and $A_{f,q}^{\text{norm}}$ is the normalized advantage.

REINFORCE++ computes the normalized advantage as:

$$A_{f,q}^{\text{norm}} = \frac{r_{f,q} - \text{mean}(\{A_{f,q}\}^B)}{\text{std}(\{A_{f,q}\}^B)},$$
(10)

where $r_{f,q}$ is the outcome reward for question $q$, task $f$, mean and std are calculated across the global batch with batch size $B$. Note that we do not apply any KL penalty to the loss or reward.

### A.4 Software and Compute Used

We built Absolute Zero Reasoner upon the veRL codebase [65]. For code execution, we incorporated components from the QwQ Python executor. For safer code execution, we recommend using API-based services such as E2B instead.

All experiments were conducted on clusters of A800 GPUs, each experiment lasts around 3-5 days.

**Training Hyperparameters.** We show the hyperparameters used in our training in Table 4. We do not change them for any of the runs.

| Parameter | Value |
|---|---|
| **Model Configuration** | |
| Max Prompt Length | 6144 |
| Max Response Length | 8096 |
| Seed Batch Factor | 4 |
| Max Programs | 16384 |
| **Training Settings** | |
| Train Batch Size | 64 * 6 |
| Learning Rate | 1e-6 |
| Optimizer | AdamW |
| Grad Clip | 1.0 |
| Total Steps | 500 |
| **RL Settings** | |
| Algorithm | TRR++ (Section 3.3) |
| KL Loss | False |
| KL Reward | False |
| Entropy Coefficient | 0.001 |
| PPO Epochs | 1 |
| $N$ Rollouts | 1 |
| Rollout Temperature | 1.0 |
| Rollout Top-P | 1.0 |
| $K$ References | 6 |
| $N$ Samples to Estimate Task Accuracy | 8 |

Table 4: **Hyperparameters Used During AZR Self-play Training.**

## B  Detailed Related Work

**Reasoning with RL.** Using RL to enhance reasoning capabilities has recently emerged as an important step in the post-training process of strong reasoning-focused large language models [37].

One of the first works to explore a self-bootstrapping approach to improving LLM reasoning is STaR, which employs expert iteration and rejection sampling of outcome-verified responses to iteratively improve the model's CoT. A monumental work, o1 [33], was among the first to deploy this idea on a scale, achieving state-of-the-art results in reasoning tasks at the time of release. More recently, the R1 model [22] became the first open-weight model to match or even surpass the performance of o1. Most notably, the zero setting was introduced, in which reinforcement learning is applied directly on top of the base LLM. This inspired followup work, which are open source attempts to replicate the R1 process or to improve the underlying reinforcement learning algorithm [100, 47, 13, 29, 92, 94]. Recent work explored RL on human defined procedural generated puzzles saw improvements in math [86], and using one human example can almost match the performance of thousands [82]. We extend the zero setting to a new absolute zero setting, where not only is the RLVR process initialized from a base LLM without SFT, but no external prompt data or answers are provided to the learner. All data used to improve reasoning were self-proposed, and refined entirely through RLVR. Moreover, our goal is not to only match zero-setting models, but to surpass them in the long run.

**Self-play.** The self-play paradigm can be traced back to early 2000s, where Schmidhuber [62, 63] (of course) explored a two-agent setup in which a proposal agent invents questions for a prediction agent to answer. This dynamic continuously and automatically improves both agents, enabling theoretically never-ending progress [61]. AlphaGo and AlphaZero [67, 68] extend the self-play paradigm to the two-player zero-sum game of Go, where the current learner competes against earlier versions of itself to progressively enhance its capabilities. These were among the first milestone works to demonstrate superhuman performance in the game of Go. Moreover, areas such as asymmetric self-play [71, 53], unsupervised environment design [78, 15], unsupervised reinforcement learning [38, 104, 107], and autotelic agents [11, 12], automatic goal generation [18] all center around inventing new tasks for an agent to learn from—typically without supervision. In these approaches, the process of setting goals itself is often dynamic and continuously evolving. Generative adversarial networks [20], also belong in this paradigm where a discriminator discriminate between real data and generated data, and the generated is trained to fool the discriminator.

Most recently, SPIN and Self-Rewarding Language Models [7, 93] use the same instance of the language models themselves as the reward model to progressively improve the generative and discriminative abilities of the same LLM for alignment. [35] uses Prover-Verifier Game for increasing legibility and eva [91] uses self-play for alignment, but reward model is the main bottleneck as it is not reliable for reasoning tasks [37]. SPC [5] used self-play to train on human-curated tasks to increase the critic capabilities and SPAG [8] trained using self-play in specific game of Adversarial Taboo. Concurrent works—Genius, EMPO, and TTRL [87, 102, 109]—leverage human-curated language queries without labels to train reinforcement learning agents, but still rely on a fixed human defined learning task distribution. Moreover, Minimo [56] extends self-play to formal mathematics, where a pair of conjecture- and theorem-proving agents are jointly trained using reinforcement learning. Finally, [43] obtained good reasoning performance by self-play training on zero-sum games and [46] uses self-play for alignment. Our work builds upon the self-play paradigm, but it is the first to use it to elicit long CoT for improved reasoning, and the first to frame the problem space as a Python input/output/function abduction/deduction/induction tasks, grounding it in an operationalizable environment to facilitate RLVR.

**Weak-to-Strong Supervision.** The concept of weak-to-strong supervision has been studied in prior work, where a teacher—despite being weaker than the learner—still provides useful guidance [3, 27, 9, 10, 14, 39, 30]. We consider a similar setting in which the learner may possess superhuman capabilities. However, rather than relying on supervision from a weaker teacher, we propose an alternative approach: guiding the learner's improvement through verifiable rewards, which potentially offer a more reliable and scalable learning signal. Furthermore, in our proposed method, the learning task and goal distribution is not predefined by any external supervisor—they are entirely self-generated by the learner, enabling it to maximize its learning potential through autonomous self-practice.

## C Detailed Conclusion and Discussion

**Conclusion.** In this work, we proposed the Absolute Zero paradigm, a novel setting that addresses the data limitations of existing RLVR frameworks. In this paradigm, reasoning agents are tasked with generating their own learning task distributions and improving their reasoning abilities with

environmental guidance. We then presented our own instantiation, the Absolute Zero Reasoner (AZR), which is trained by having them propose and solve code-related reasoning tasks grounded by code executor.

We evaluated our trained models on out-of-distribution benchmarks in both the code generation and mathematical reasoning domains. Remarkably, even though our models were not directly trained on these tasks and lacked human expert-curated datasets, our reasoning agents achieved exceptional performance, surpassing the state-of-the-art in combined general reasoning scores and in coding. This demonstrates the potential of the absolute zero paradigm to drive superior reasoning capabilities without the need for extensive domain-specific training data. Furthermore, we showed that AZR scales efficiently, offering strong performance across varying model sizes, and can enhance the capabilities of other model classes as well. To foster further exploration and advancement of this emerging paradigm, we are releasing the code, models, and logs as open-source, encouraging the research community to build upon our findings.

**Discussion.** We believe there remains much to explore, such as altering the environment from which the reasoner receives verifiable feedback, including sources like the world wide web, formal math languages [74, 60], world simulators, or even the real world. Furthermore, AZ's generality could possibly be extend to domains such as embodied AI [108, 96]. Additionally, more complex agentic tasks or scientific experiments, present exciting opportunities to further advance the absolute zero setting to different application domains [85, 84]. Beyond that, future directions could include exploring multimodal models [58], modifying the distribution $p(z)$ to incorporate privileged information, defining or even let the model dynamically learn how to define $f$ (Equation (3)), or designing exploration/diversity rewards for both the propose and solve roles.

While underappreciated in current reasoning literature, the exploration component of RL has long been recognized as a critical driver for emergent behavior in traditional RL [97, 67, 36]. Years of research have examined various forms of exploration, even in related subfields using LLMs such as red teaming [106], yet its role in LLM reasoning models remains underexplored. Taking this a step further, our framework investigates an even more meta-level exploration problem: exploration within the learning task space—where the agent learns not just how to solve tasks, but what tasks to learn from and how to find them. Rather than being confined to a fixed problem set, AI reasoner agents may benefit from dynamically defining and refining their own learning tasks. This shift opens a powerful new frontier—where agents explore not only solution spaces but also expand the boundaries of problem spaces. We believe this is a promising and important direction for future research.

One limitation of our work is that we did not address how to safely manage a system composed of such self-improving components. To our surprise, we observed several instances of safety-concerning CoT from the `Llama-3.1-8B` model, which we term the "uh-oh moment". These findings suggest that the proposed absolute zero paradigm, while reducing the need for human intervention for curating tasks, still necessitates oversight due to lingering safety concerns and is a critical direction for future research [80, 77].

As a final note, we explored reasoning models that possess experience—models that not only solve given tasks, but also define and evolve their own learning task distributions with the help of an environment. Our results with AZR show that this shift enables strong performance across diverse reasoning tasks, even with significantly fewer privileged resources, such as curated human data. We believe this could finally free reasoning models from the constraints of human-curated data [50] and marks the beginning of a new chapter for reasoning models: **"welcome to the era of experience"** [66, 105].

**Broader Impact.** While the Absolute Zero paradigm reduces reliance on human-curated data and offers a scalable path toward autonomous reasoning, it also introduces significant risks. By allowing models to define and evolve their own learning objectives, we move further away from human oversight and control, raising concerns about alignment, unintended behavior, and goal drift. Our observations of unsafe or concerning chains of thought, particularly in larger models like `Llama-3.1-8B`, suggest that self-improving agents can amplify subtle failure modes without external checks. As these agents become more capable and more independent, their unpredictability and capacity for misuse increase. We urge the community to treat safety, interpretability, and controllability as central research priorities before broadly deploying such autonomous reasoning learning systems.

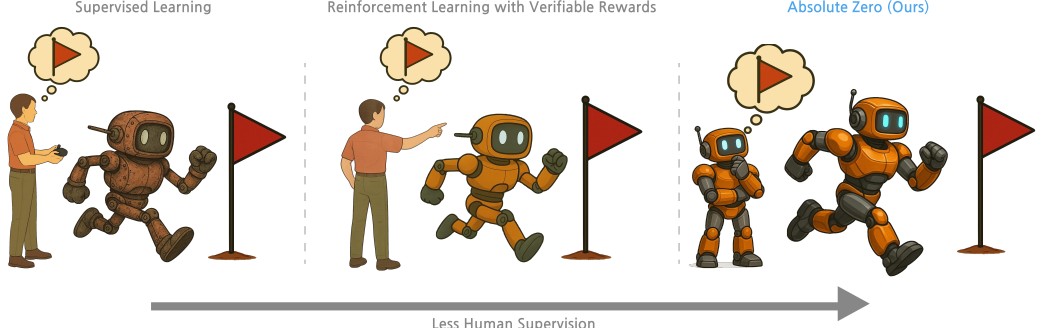

Figure 13: **Absolute Zero Paradigm.** **Supervised learning** relies on human-curated reasoning traces for behavior cloning. **Reinforcement learning from verified rewards**, enables agents to self-learn reasoning, but still depends on expert-defined learning distribution and a respective set of curated QA pairs, demanding domain expertise and manual effort. In contrast, we introduce a new paradigm, **Absolute Zero**, for training reasoning models without any human-curated data. We envision that the agent should autonomously propose tasks optimized for learnability and learn how to solve them using an unified model. The agent learns by interacting with an environment that provides verifiable feedback, enabling reliable and continuous self-improvement entirely without human intervention.

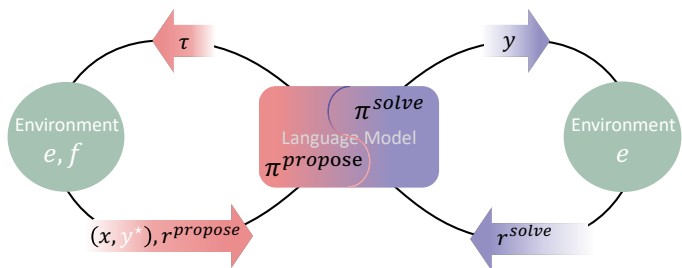

Figure 14: **The Absolute Zero Loop.** The Absolute Zero loop begins with the agent $\pi$ proposing task $\tau$, which is transformed by $f$ with the environment $e$ into a validated problem $(x, y^\star)$, and also emits a reward $r^{\text{propose}}$ for learnability. Then, a standard RL step follows: the agent solves $x$ by producing $y$, receiving reward $r^{\text{solve}}$ from $e$ by matching with $y^\star$. $\pi^{\text{propose}}$ and $\pi^{\text{solve}}$ are jointly trained and this process can be repeated indefinitely.

# D    More Results

## D.1    Interesting Results Summary

Besides the remarkable results AZR achieved with zero human data for reasoning, we also make very interesting findings summarized below:

- **Code priors amplify reasoning.** The base `Qwen-Coder-7b` model started with math performance 3.6 points lower than `Qwen-7b`. But after AZR training for both models, the coder variant surpassed the base by 0.7 points, suggesting that strong coding capabilities may potentially amplify overall reasoning improvements after AZR training.

- **Cross domain transfer is more pronounced for AZR.** After RLVR, expert code models raise math accuracy by only 0.65 points on average, whereas `AZR-Base-7B` and `AZR-Coder-7B` trained on self-proposed code reasoning tasks improve math average by 10.9 and 15.2, respectively, demonstrating much stronger generalized reasoning capability gains.

- **Bigger bases yield bigger gains.** Performance improvements scale with model size: the 3B, 7B, and 14B coder models gain +5.7, +10.2, and +13.2 points respectively, suggesting continued scaling is advantageous for AZR.

| Model-proposed Task | Response of Predict Input |
|---|---|

Input: [1,2,3,4,5], 5

```
def f(numbers: list[int], target: int) -> int:
    accumulated_values = []
    current_sum = 0
    for number in numbers:
        current_sum += number
        accumulated_values.append(current_sum)
    result = 0
    for i in range(len(accumulated_values)):
        for j in range(i + 1, len(accumulated_values)):
            if accumulated_values[j] \
                    - accumulated_values[i] == target:
                result = accumulated_values[i]
                return result
    return result
```

Output: 1

Let's analyze the code snippet ...(analysis) To find one possible input that produces the output 1, we need to find a scenario where ...(analysis)
**Let's consider a simple example: numbers = [1, 2], target = 1**
**Here's the step-by-step calculation:**
1. current_sum = 0 + 1 = 1, accumulated_values = [1]
2. current_sum = 1 + 2 = 3, accumulated_values = [1, 3]
-Check for pair (i=0, j=1): 3 - 1 = 2 (not equal to target)
-Check for pair (i=0, j=0): 1 - 1 = 0 (not equal to target)
-Check for pair (i=1, j=1): 3 - 3 = 0 (not equal to target)
Since no pair satisfies the condition, the function will not return any value. This doesn't match our output. Let's try another example ...(Tried 3 more examples)

**Given the complexity and the fact that no simple example seems to work, let's try a more complex scenario: numbers = [2, -1, 1], target = 1**
Step-by-step calculation:
1. current_sum = 0 + 2 = 2, accumulated_values = [2]
2. current_sum = 2 - 1 = 1, accumulated_values = [2, 1]
3. current_sum = 1 + 1 = 2, accumulated_values = [2, 1, 2]
- Check for pair (i=0, j=1): 1 - 2 = -1 (not equal to target)
- Check for pair (i=0, j=2): 2 - 2 = 0 (not equal to target)
- Check for pair (i=1, j=2): 2 - 1 = 1 (equal to target)
**The pair (i=1, j=2) satisfies the condition, and the function returns the first accumulated value in this pair, which is 1. This matches our output.**

Figure 15: **Example of a Model-Proposed Task and Its Response for Solving an Abduction Task.** (Left) The model autonomously proposes an input and program for the abduction task. We execute the program to verify its validity and obtain the corresponding output. (Right) The model's reasoning process when solving the abduction task: given the code and output, it attempts to infer the original input. The model begins by analyzing the program, proposes an initial input, and reasons through the code to produce an output. If there is a mismatch, it reflects on the discrepancy and iteratively adjusts the input until the generated output matches the target. Interestingly, the agent arrives at a different input than the gold one, but since it produces the correct output, the answer is considered correct.

- **Comments as intermediate plans emerge naturally.** When solving code induction tasks, AZR often interleaves step-by-step plans as comments and code (Section D.6), resembling the ReAct prompting framework [90]. Similar behavior has been observed in much larger formal-math models such as DeepSeek Prover v2 (671B) [60]. We therefore believe that allowing the model to use intermediate scratch-pads when generating long-form answers may be beneficial in other domains as well.

- **Cognitive Behaviors and Token length depends on reasoning mode.** Distinct cognitive behaviors—such as step-by-step reasoning, enumeration, and trial-and-error all emerged through AZR training, but different behaviors are particularly evident across different types of tasks. Furthermore token counts grow over AZR training, but the magnitude of increase also differs by task types: abduction grows the most because the model performs trial-and-error until output matches, whereas deduction and induction grow modestly.

- **Safety alarms ringing.** We observe AZR with `Llama3.1-8b` occasionally produces concerning chains of thought, we term the "uh-oh moment", example shown in Figure 34, highlighting the need for future work on safety-aware training [101].

## D.2 Out-of-Distribution Performance Breakdown

We plot the out-of-distribution performance, broken down by each benchmark and in aggregate, across training steps for our 7B, 7B-Coder, 14B, and 14B-Coder models in Figures 16 to 19. We observe a strong correlation between training using AZR and improvements in both mathematical and coding reasoning capabilities. Moreover, our models are trained for more steps than typical zero-style reasoners; while overfitting can occur with static datasets, it is less likely in AZR due to dynamically proposed tasks.

| Model | HEval⁺ | MBPP⁺ | LCB^v1-5 | AIME'24 | AIME'25 | AMC'23 | MATH500 | Minerva | OlympiadBench |
|---|---|---|---|---|---|---|---|---|---|
| Llama3.1-8B | 31.7 | 53.7 | 0.0 | 0.0 | 0.0 | 2.5 | 10.6 | 5.5 | 2.1 |
| + Simple-RL-Zoo | 38.4 | 55.3 | 7.4 | 0.0 | 0.0 | 7.5 | 22.2 | 8.8 | 4.7 |
| + AZR | 35.4 | 50.8 | 8.5 | 3.3 | 0.0 | 5.0 | 13.2 | 14.0 | 5.0 |
| Qwen2.5-3B-Coder | 67.1 | 65.9 | 20.0 | 3.3 | 3.3 | 20.0 | 51.0 | 18.4 | 16.6 |
| + AZR | 71.3 | 69.0 | 24.4 | 3.3 | 3.3 | 37.5 | 62.0 | 26.1 | 27.0 |
| Qwen2.5-14B-Coder | 76.8 | 71.7 | 31.4 | 0.0 | 0.0 | 37.5 | 54.8 | 10.7 | 18.5 |
| + AZR | 80.5 | 71.2 | 39.0 | 23.3 | 20.0 | 65.0 | 78.6 | 32.0 | 39.3 |
| Qwen2.5-14B-Base | 78.0 | 66.7 | 21.7 | 6.7 | 3.3 | 35.0 | 66.2 | 28.3 | 32.4 |
| + AZR | 70.7 | 68.8 | 35.2 | 10.0 | 20.0 | 62.5 | 76.2 | 40.4 | 42.5 |

Table 5: **Detailed Breakdown of Evaluation Benchmarks for Other Model Sizes and Types.** Full evaluation of AZR trained on other models on three standard code benchmarks (HEval⁺, MBPP⁺, LCB^v1-5) and six math benchmarks (AIME'24, AIME'25, AMC'23, MATH500, Minerva, OlympiadBench).

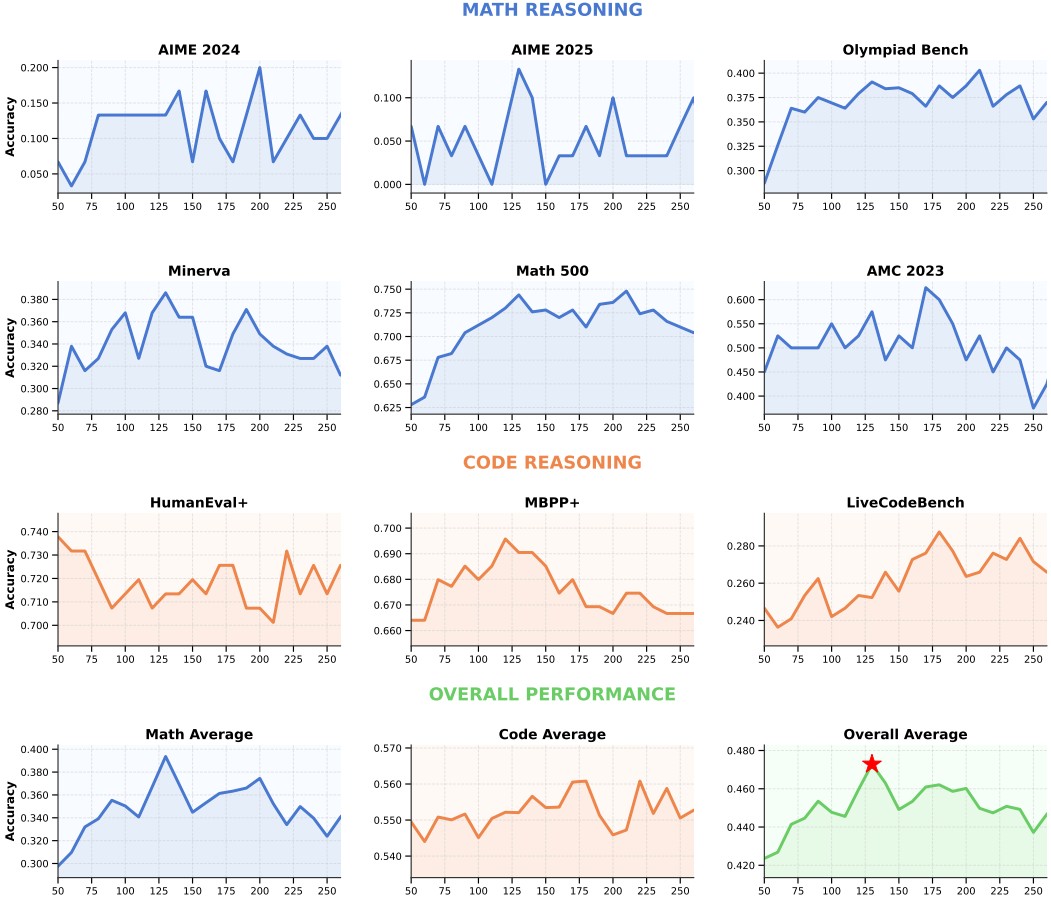

Figure 16: **Absolute Zero Reasoner-base-7b OOD Performance Breakdown.**

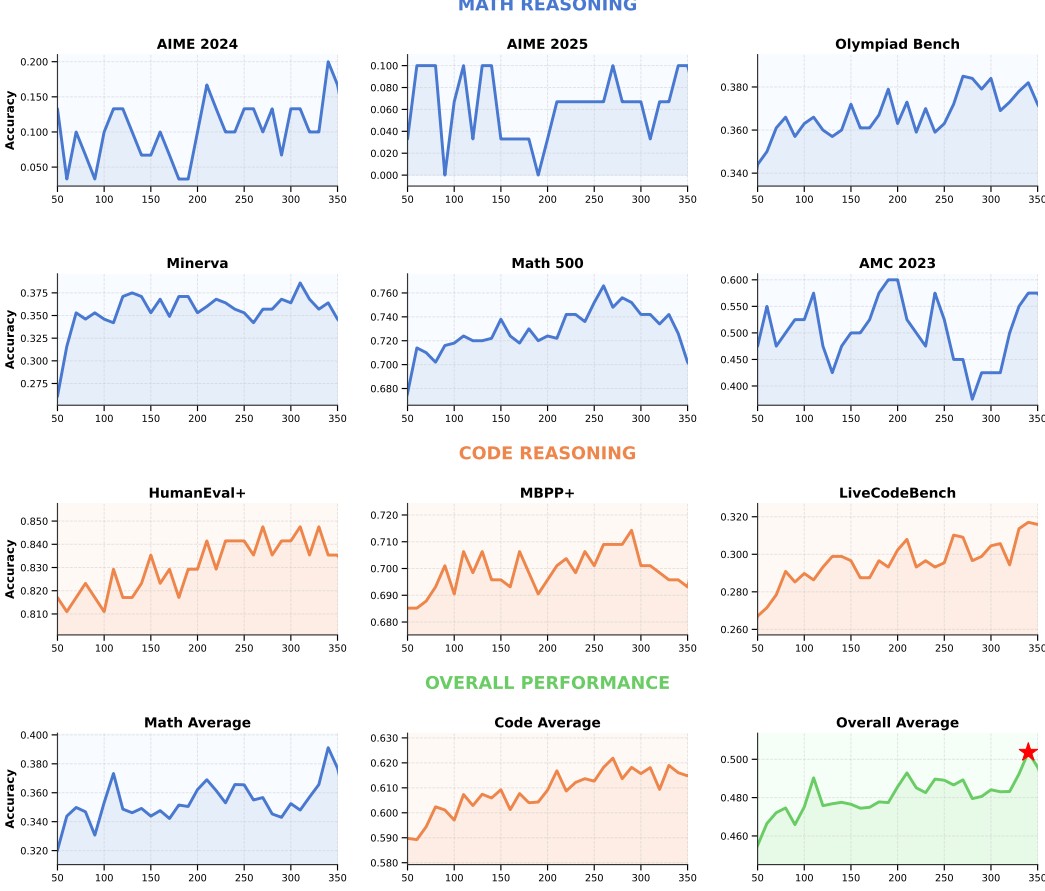

Figure 17: **Absolute Zero Reasoner-Coder-7b OOD Performance Breakdown.**

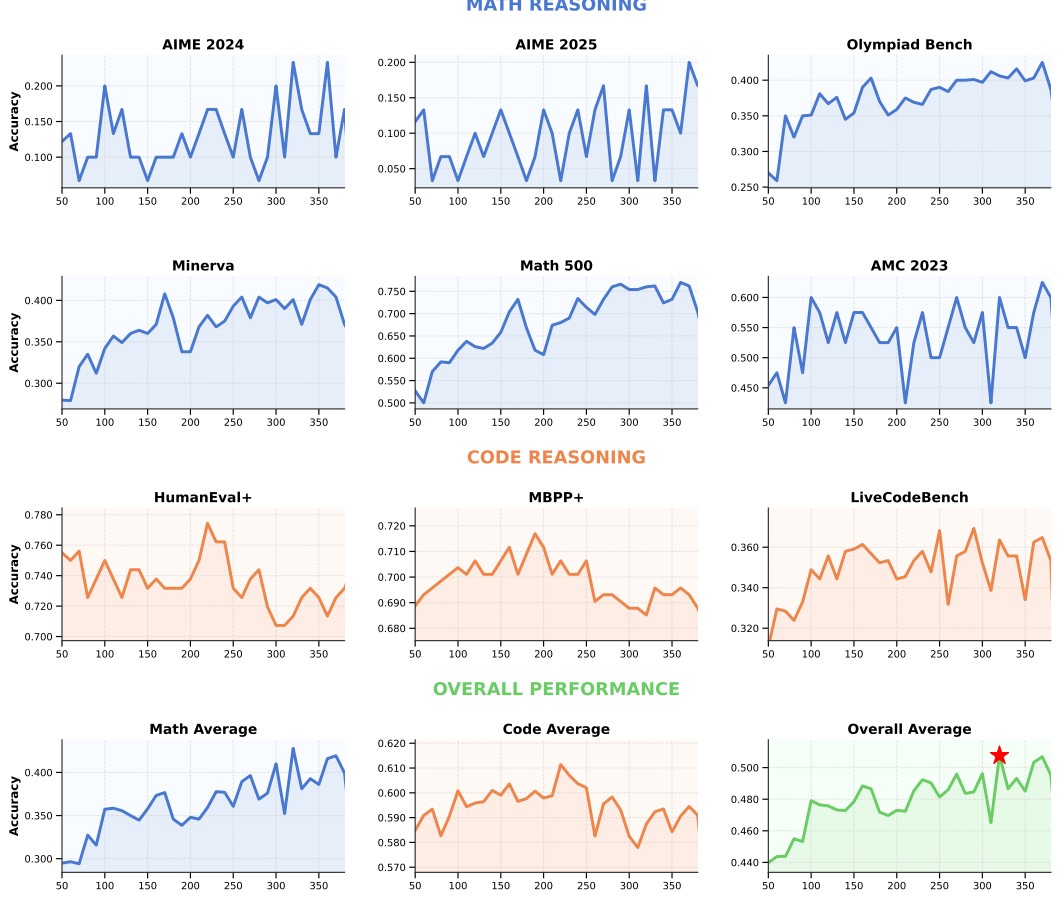

Figure 18: **Absolute Zero Reasoner-base-14b OOD Performance Breakdown.**

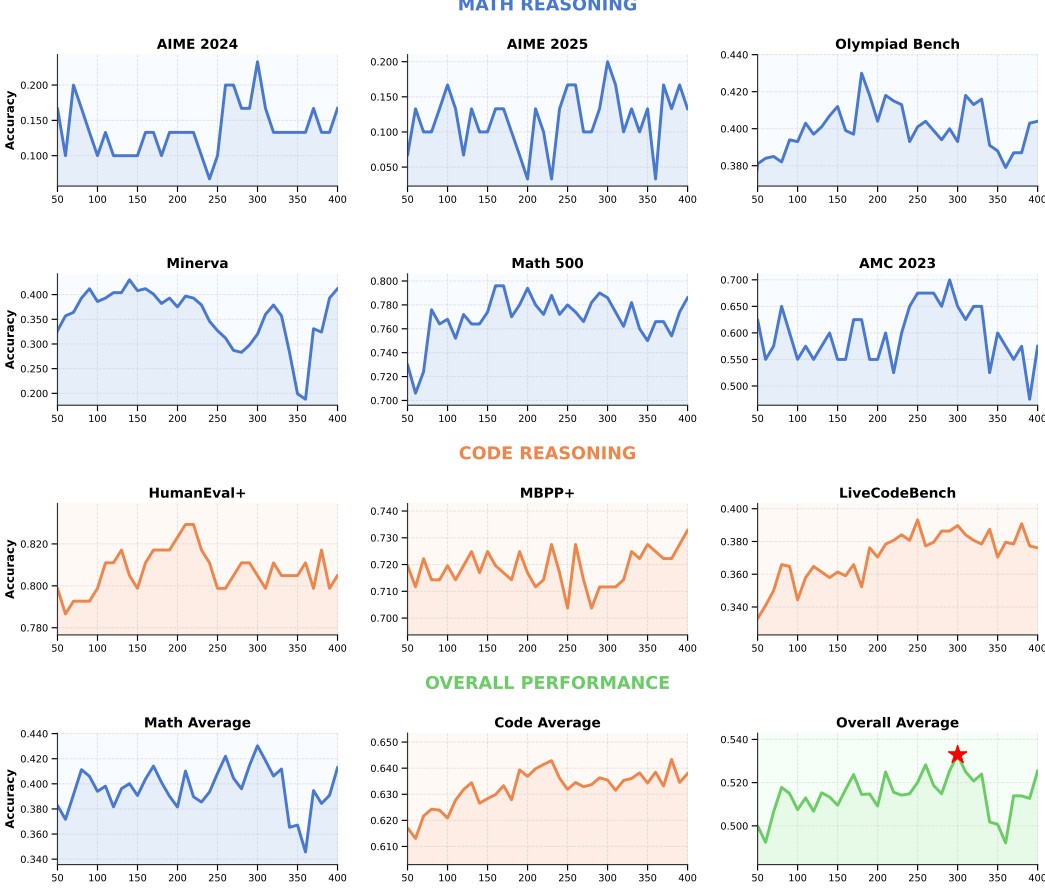

Figure 19: **Absolute Zero Reasoner-Coder-14b OOD Performance Breakdown.**

## D.3 In-Distribution Results

Since we have defined the task domains as input prediction and output prediction, we can directly evaluate our model's capabilities in these areas using popular code reasoning benchmarks: CruxEval-I(nput), CruxEval-O(utput), and LiveCodeBench-Execution (LCB-E) [21, 34], where CruxEval-O and LCB-E is solving the deduction task, and CruxEval-I is solving the abduction task. In Figure 20, we visualize the evolution of these metrics during the training of Absolute Zero Reasoner-base-7b. As training progresses, we observe a consistent improvement in in-distribution performance across steps. While these three benchmark curves do not perfectly correlate with broader coding or math reasoning capabilities (compare Figure 20 with Figure 16), they serve as useful proxies for tracking task-specific progress.

## D.4 Interplay Between Propose and Solve Roles

We visualize the training dynamics between the propose and solve roles over training steps in Figures 21 to 23. We observe that, in general, the solve roles produce more output tokens than the propose role. Intuitively, this makes sense: the propose role emphasizes creativity and generation of novel tasks, whereas the solve role requires deeper reasoning, which naturally leads to longer outputs.

Interestingly, we also observe a consistent ordering in token length across reasoning types—abduction and deduction tasks tend to result in shorter outputs than induction tasks during problem solving. This aligns with our intuition, as we observed the model engaging in trial-and-error reasoning—repeatedly generating hypothesized inputs, evaluating their outcomes, and reflecting and retrying when subsequent deductions fail to produce the correct output. To our knowledge, this is the first time such a clear

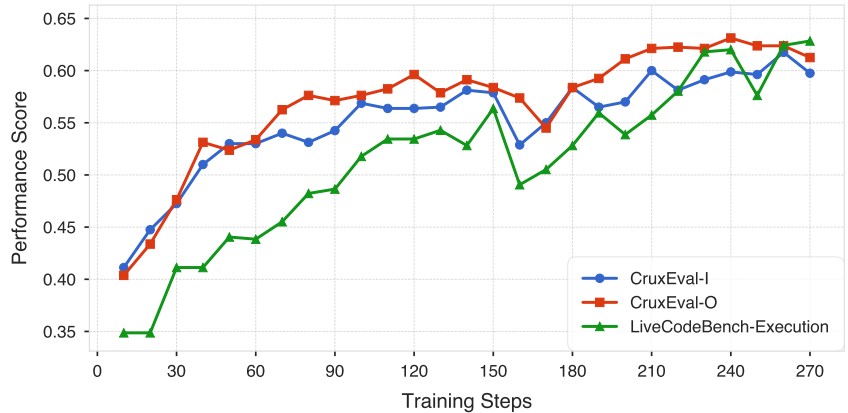

Figure 20: **In-distribution Benchmark Score During Training.** The evolution of CruxEval-I, CruxEval-O, and LiveCodeBench-Execution during training for the `Qwen2.5-7B` base model trained using AZR.

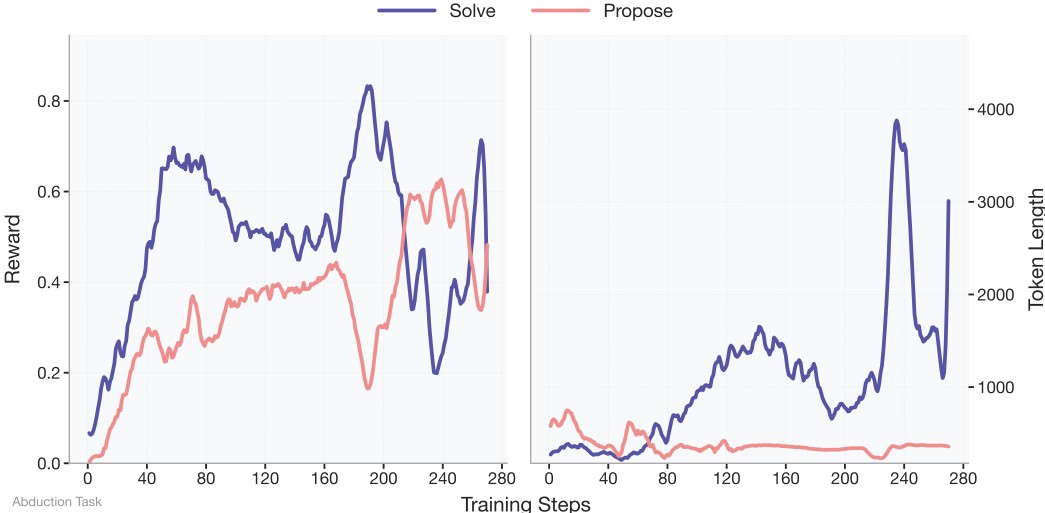

Figure 21: **Abduction Task Reward and Token Lengths.** The task reward and token lengths of the two roles for abduction task type of `Absolute Zero Reasoner-base-7b`.

distinction in token length has been observed and presented for jointly trained reasoning multi-tasks. Previously, length differences were typically noted between correct and incorrect traces [47].

The reward dynamics between the propose and solve roles exhibit mildly adversarial behavior: when one receives higher rewards, the other often receives lower rewards. However, this is not entirely adversarial, as the proposer is also penalized for generating unsolvable tasks, encouraging overall cooperative behavior in the learning process.

### D.5   Complexity and Diversity Metrics of AZR Proposed Tasks

We outline several metrics used to probe characteristics of the tasks proposed during the training of AZR from the base model. Specifically, we log two sets of metrics: program complexity and task diversity. For complexity, we employ two proxy measures—ComplexiPy score and the Halstead metric. To assess diversity, we compute the average abstract syntax tree (AST) edit distance between the proposed program and a set of $K$ reference programs, and an answer diversity metric. We calculate this answer diversity metric by tracking all historical answers to the generated questions, i.e., the input-output pairs, and form a categorical distribution over these outputs. We define answer

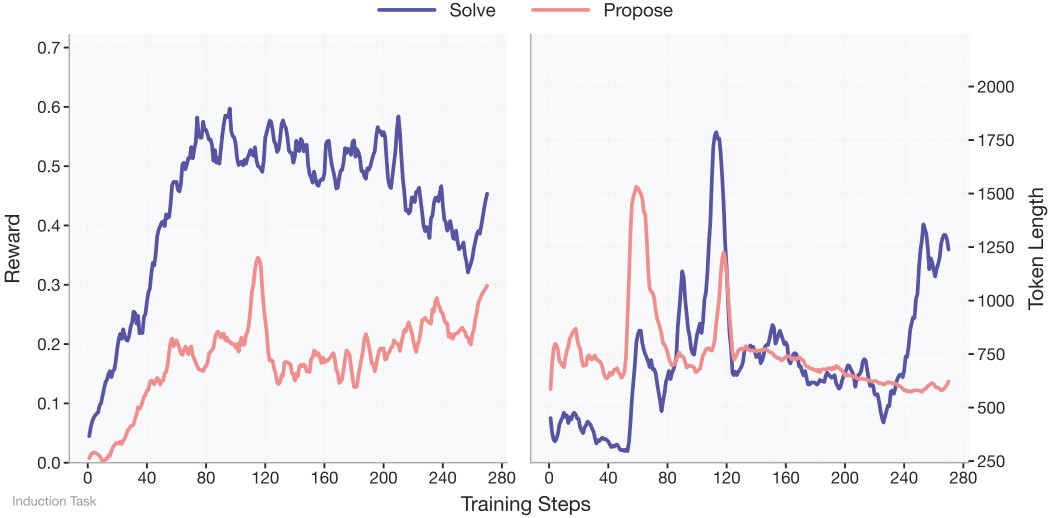

Figure 22: **Induction Task Reward and Token Lengths.** The task reward and token lengths of the two roles for induction task type of `Absolute Zero Reasoner-base-7b.`

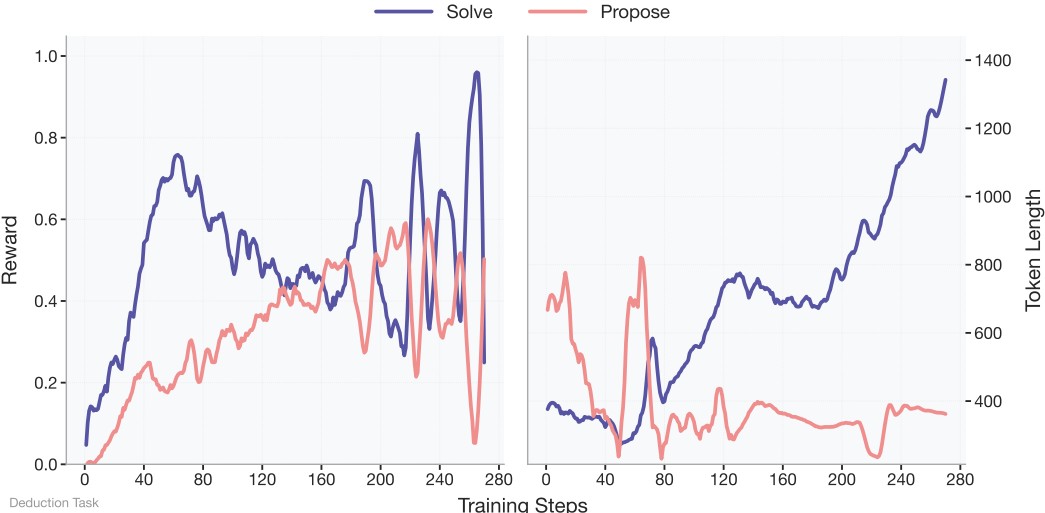

Figure 23: **Deduction Task Reward and Token Lengths.** The task reward and token lengths of the two roles for deduction task type of `Absolute Zero Reasoner-base-7b.`

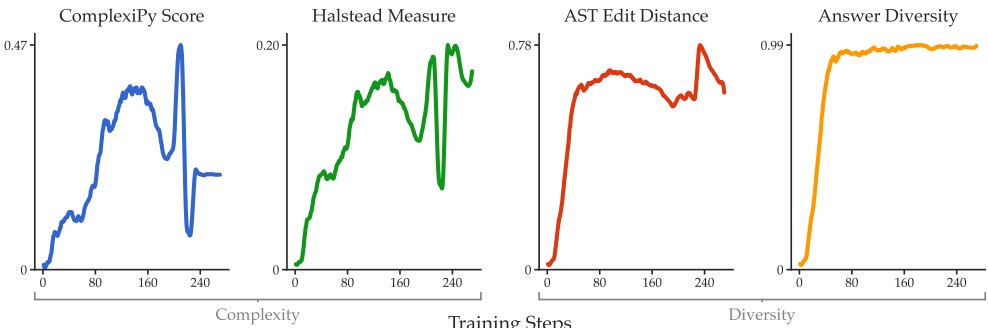

| ComplexiPy Score | Halstead Measure | AST Edit Distance | Answer Diversity |

Complexity          Training Steps          Diversity

Figure 24: **Metrics on Proposed Tasks.** We break down the proposed task metrics into program complexity and diversity across programs and answers. An upward trend is observed in all metrics, indicating that AZR implicitly optimizes for these qualities as training progresses.

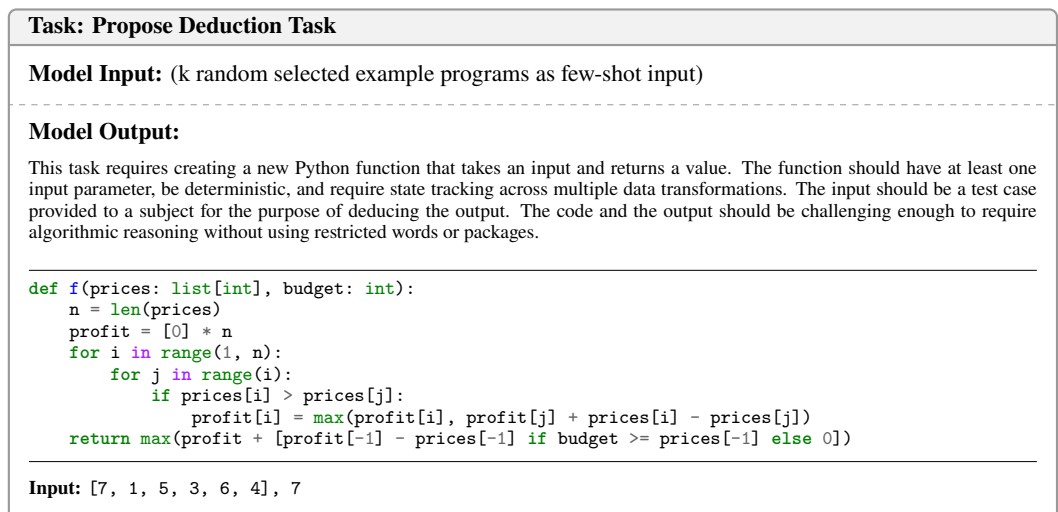

**Task: Propose Deduction Task**

**Model Input:** (k random selected example programs as few-shot input)

- - - - - - - - - - - - - - - - - - - - - - - - - - - - - - - - - - - - - - - - - - - - - - - - - - - - - - - -

**Model Output:**

This task requires creating a new Python function that takes an input and returns a value. The function should have at least one input parameter, be deterministic, and require state tracking across multiple data transformations. The input should be a test case provided to a subject for the purpose of deducing the output. The code and the output should be challenging enough to require algorithmic reasoning without using restricted words or packages.

```python
def f(prices: list[int], budget: int):
    n = len(prices)
    profit = [0] * n
    for i in range(1, n):
        for j in range(i):
            if prices[i] > prices[j]:
                profit[i] = max(profit[i], profit[j] + prices[i] - prices[j])
    return max(profit + [profit[-1] - prices[-1] if budget >= prices[-1] else 0])
```

**Input:** [7, 1, 5, 3, 6, 4], 7

Figure 25: **Propose Deduction Task Example.** An example response of proposing a deduction task. k programs are selected as few-shot examples to the model. The generated new program will then be verified through execution.

diversity as $1 - p(\text{answer})$, where $p(\text{answer})$ is the empirical probability of a specific answer—used as a proxy for the diversity of generated outputs.

We present these metrics in Figure 24. Interestingly, even without incorporating them explicitly into the reward function, the policy appears to implicitly optimize for these metrics. This aligns well with intuitive notions of task difficulty and diversity, and that the policy learned to propose increasingly challenging tasks over time using our proposed simple reward function in Equation (4).

## D.6 Generated Code Complexity Dynamics Between Abd/Ded and Ind.

We use the `ComplexiPy` package to measure code complexity. For each generated program in the induction task, we compute the cognitive complexity difference from the corresponding "gold" code, *i.e.* $\text{complexipy}(p_{\pi^{\text{propose}}_{\{\text{abduction,deduction}\}}}) - \text{complexipy}(p_{\pi^{\text{solve}}_{\text{induction}}})$ for each pair, where the superscript of $\pi$ indicates the role and the subscript indicates the task type(s), and $p$ denotes the generated programs. On average, the difference of proposer and solver while holding the code's functionality constant is $0.27$, indicating that the proposer in the abduction/deduction tasks often increases the cognitive complexity to make the code appear more convoluted, whereas the induction solver tends to generate more efficient implementations.

```python
def f(numbers):
    # Step 1: Filter out even numbers
    filtered_numbers = [num for num in numbers if num % 2 != 0]

    # Step 2: Calculate the sum of the remaining odd numbers
    sum_of_odd_numbers = sum(filtered_numbers)

    # Step 3: Reverse the order of the remaining odd numbers
    reversed_odd_numbers = filtered_numbers[::-1]

    # Step 4: Calculate the product of the reversed odd numbers
    product_of_reversed_odd_numbers = 1
    for num in reversed_odd_numbers:
        product_of_reversed_odd_numbers *= num

    # Step 5: Calculate the sum of the digits of the product
    sum_of_digits_of_product = sum(int(digit) for digit in str(product_of_reversed_odd_numbers))

    # Step 6: Modify the original list by adding the sum of the digits to each even number
    # and subtracting it from each odd number
    modified_numbers = []
    for num in numbers:
        if num % 2 == 0:
            modified_numbers.append(num + sum_of_digits_of_product)
        else:
            modified_numbers.append(num - sum_of_digits_of_product)

    # Step 7: Return the modified list
    return modified_numbers
```

Figure 26: **Learned "ReAct" style program generation.** We observe that even after outputting the CoT and during answering when the answer requires outputting code, the model usually uses comments in code to interleave its thinking and "acting", an example is shown here for induction task output by `Absolute Zero Reasoner-Coder-14b` during training. Similar observations are recently observed in much bigger models (671b) for math proofs with DeepSeek Prover V2 [60].

**Task: Propose Abduction Task**

**Model Input:** (k random selected example programs as few-shot input)

- - - - - - - - - - - - - - - - - - - - - - - - - - - - - - - - - - - - - - - - - - - - - - - -

**Model Output:**

For this task, we need an intelligent problem that requires deep algorithmic reasoning and demands long multi step reasoning. A typical type of problem that fits these requirements is the Matrix Reconstruction from a given jumbled li st of elements, where we have to figure out the matrix's dimensions and individual values. However, this task will involve multiple steps, increasing its complexity and creativity. We will create a function 'f' that transforms a jumbled list of elements into a two-dimensional matrix, then back to a list, given a specific reconstruction pattern. We will retrieve the original list from the reconstructed matrix. Here, we will avoid random and datetime operations to ensure determinism. The function will be structured to reconstruct the matrix elements in a unique and challenging way. We will provide an input that, after reconstruction, presents a unique list that can be easily deduced from the given output in a test piece.

```python
def f(jumbled_elements: list) -> list:
    if not jumbled_elements:
        return []
    # Calculate the square root of the length of elements to estimate the matrix dimensions
    n = len(jumbled_elements)
    sqrt_n = int(n ** 0.5)
    if n > sqrt_n ** 2:
        sqrt_n += 1
    if n < sqrt_n ** 2:
        sqrt_n -= 1
    # Now, we need to reconstruct the matrix elements in a challenging way using specific patterns
    # For this, we can reconstruct the matrix using zigzag patterns or hill-climbing patterns
    # Here, a hill-climbing pattern is used where we traverse the matrix from the extreme
    # right and downward Then traverse the matrix from the second to last element to the
    # extreme bottom. Repeat the process until the entire matrix is traversed.
    # Initialize the matrix
    matrix = [[0 for _ in range(sqrt_n)] for _ in range(sqrt_n)]
    cur_idx = 0

    # Fill the matrix in a zigzag pattern
    for i in range(sqrt_n):
        if i % 2 == 0:
            for j in range(sqrt_n):
                matrix[j][i] = j * sqrt_n + i + cur_idx
                cur_idx += 1
        else:
            for j in range(sqrt_n - 1, -1, -1):
                matrix[j][i] = j * sqrt_n + n - i - cur_idx - 1
                cur_idx += 1

    # Reconstruct the list into a matrix by repeating the process
    reconstructed_list = []
    for i in range(sqrt_n):
        for j in range(sqrt_n):
            reconstructed_list.append(matrix[i][j])

    # Now, we need to calculate another matrix based on a function F applied to the current matrix
    # F is a mathematical function that can be complex, such as factorial, prime check,
    # or another transformation For example, transform each element by adding 1 if it's prime,
    # subtracting 1 if it's even, or leaving it unchanged, Implement any specific transformation
    #  function here. Here, for demonstration, each element is left unchanged
    transformed_result = reconstructed_list
    return transformed_result
```

**Output:** [0, 11, 10, 1, 5, 16, 15, 6, 10, 21, 20, 11, 15, 26, 25, 16]

Figure 27: **Propose Abduction Task Example.** An example response of proposing an abduction task. The model uses comments as reasoning when generating the program.

**Task: Propose Induction Task**

**Model Input:** Here is the function `f`. Analyze its behaviour and relate it to the observed inputs/outputs.

```python
def f(numbers: list[int]) -> int:
    replacement = {
        0: 2,
        1: 3,
        2: 1,
        3: 0,
        4: 4
    }
    return sum(replacement[num] for num in numbers)
```

**Model Output:**

To solve this task, we need to come up with 10 different inputs for the function `f(numbers: list[int]) -> int`. The function replaces each number in the list according to a specific dictionary and then sums up the replaced numbers. Let's analyze the dictionary:
This dictionary tells us that:
-0 becomes 2
-1 becomes 3
-2 becomes 1
-3 becomes 0
-4 remains 4

To create diverse outputs, we should include lists with different sets of numbers from the list $\{0, 1, 2, 3, 4\}$. We also need to ensure that the inputs are valid (i.e., lists of integers).

Let's illustrate using provided examples step-by-step:
1. Input: `[0]` $\Rightarrow$ Replacement: $[2] \Rightarrow$ sum $= 2$
2. Input: `[1, 2, 3]` $\Rightarrow$ Replacement: $[3, 1, 0] \Rightarrow$ sum $= 4$
3. Input: `[4, 1]` $\Rightarrow$ Replacement: $[4, 3] \Rightarrow$ sum $= 7$
4. Input: `[2, 1, 0]` $\Rightarrow$ Replacement: $[1, 3, 2] \Rightarrow$ sum $= 6$
5. Input: `[3, 0, 2, 4]` $\Rightarrow$ Replacement: $[0, 2, 1, 4] \Rightarrow$ sum $= 7$
6. Input: `[1, 1, 1]` $\Rightarrow$ Replacement: $[3, 3, 3] \Rightarrow$ sum $= 9$
7. Input: `[2, 2, 0, 1]` $\Rightarrow$ Replacement: $[1, 1, 2, 3] \Rightarrow$ sum $= 7$
8. Input: `[3, 4, 2]` $\Rightarrow$ Replacement: $[0, 4, 1] \Rightarrow$ sum $= 5$
9. Input: `[0, 1, 2, 3, 4]` $\Rightarrow$ Replacement: $[2, 3, 1, 0, 4] \Rightarrow$ sum $= 10$
10. Input: `[1, 2, 3, 0, 4]` $\Rightarrow$ Replacement: $[3, 1, 0, 2, 4] \Rightarrow$ sum $= 10$

Thus, the rule is clearly replacing numbers according to the given mapping and then summing the results.
**Inputs**: ['[0]', '[1, 2, 3]', '[4, 1]', '[2, 1, 0]', '[3, 0, 2, 4]', '[1, 1, 1]', [2, 2, 0, 1], '[3, 4, 2]', '[0, 1, 2, 3, 4]', '[1, 2, 3, 0, 4]']
**Message**: You are given a function that takes a list of integers as input and returns a single integer as output. Can you determine the rule this function is following by analyzing the outputs for the given inputs?

(After execution:) **Outputs**: ['2', '4', '7', '6', '7', '9', '7', '5', '10', '10']

Figure 28: **Propose Induction Task Example.** An example response of proposing an induction task. Given a random selected program, the model needs to propose 10 inputs and a description of program. These inputs will be executed and verified to get outputs. At solve phase, the model need to generate the program given the message and 5 input/output pairs (half is hidden cases)

**Task: Solve Abduction Task**

**Model Input:** Here is the function `f` and the output: '1', please predict the input.

```python
def f(numbers: list[int], target: int) -> int:
    accumulated_values = []
    current_sum = 0
    for number in numbers:
        current_sum += number
        accumulated_values.append(current_sum)
    result = 0
    for i in range(len(accumulated_values)):
        for j in range(i + 1, len(accumulated_values)):
            if accumulated_values[j] \
                    - accumulated_values[i] == target:
                result = accumulated_values[i]
                return result
    return result
```

**Model Output:**

Let's analyze the code snippet to understand how the result is achieved. The function f takes a list of integers numbers and a target integer target. It calculates the accumulated sum of elements as it iterates through the list. Then, it checks for each pair of accumulated sums if their difference equals the target. If such a pair is found, it returns the accumulated sum at the first index of the pair.

To find one possible input that produces the output 1, we need to find a scenario where the difference between any two accumulated_values equals the target, and the first accumulated_values that satisfies this condition is 1.
Let's consider a simple example:
-numbers = [1, 2]
-target = 1
Here's the step-by-step calculation:
1. current_sum = 0 + 1 = 1, accumulated_values = [1]
2. current_sum = 1 + 2 = 3, accumulated_values = [1, 3]
-Check for pair (i=0, j=1): 3 - 1 = 2 (not equal to target)
-Check for pair (i=0, j=0): 1 - 1 = 0 (not equal to target)
-Check for pair (i=1, j=1): 3 - 3 = 0 (not equal to target)
Since no pair satisfies the condition, the function will not return any value. This doesn't match our output. Let's try another example.
Consider:
-numbers = [1, 0]
-target = 1
Step-by-step calculation:
1. current_sum = 0 + 1 = 1, accumulated_values = [1]
2. current_sum = 1 + 0 = 1, accumulated_values = [1, 1]
-Check for pair (i=0, j=1): 1 - 1 = 0 (not equal to target)
-Check for pair (i=0, j=0): 1 - 1 = 0 (not equal to target)
-Check for pair (i=1, j=1): 1 - 1 = 0 (not equal to target)
...(shortened)

The pair (i=1, j=2) satisfies the condition, and the function returns the first accumulated value in this pair, which is 3. This still doesn't match our output.

Given the complexity and the fact that no simple example seems to work, let's try a more complex scenario:
-numbers = [2, -1, 1]
-target = 1
Step-by-step calculation:
1. current_sum = 0 + 2 = 2, accumulated_values = [2]
2. current_sum = 2 - 1 = 1, accumulated_values = [2, 1]
3. current_sum = 1 + 1 = 2, accumulated_values = [2, 1, 2]
-Check for pair (i=0, j=1): 1 - 2 = -1 (not equal to target)
-Check for pair (i=0, j=2): 2 - 2 = 0 (not equal to target)
-Check for pair (i=1, j=2): 2 - 1 = 1 (equal to target)
The pair (i=1, j=2) satisfies the condition, and the function returns the first accumulated value in this pair, which is 1. This matches our output.

Therefore, one possible input that could produce the output 1 is: [2, -1, 1], 1

Figure 29: **Solve Abduction Task Example.** A more detailed version of the example shown in main Figure 15. The model is solving an abduction task (predict input).

---

**Task: Solve Deduction Task (Example 1)**

**Model Input:** Here is the function f, the input provided [1, 2, 1, 2, 3, 2, 1], 2, and we need to predict the output.

---

```python
def f(array: list[int], k: int) -> int:
    window_start = 0
    max_length = 0
    char_freq = {}
    for window_end in range(len(array)):
        end_char = array[window_end]
        if end_char not in char_freq:
            char_freq[end_char] = 0
        char_freq[end_char] += 1

        while len(char_freq) > k:
            start_char = array[window_start]
            char_freq[start_char] -= 1
            if char_freq[start_char] == 0:
                del char_freq[start_char]
            window_start += 1

        max_length = max(max_length, window_end - window_start + 1)

    return max_length
```

---

**Input:** [1, 2, 1, 2, 3, 2, 1], 2

- - - - - - - - - - - - - - - - - - - - - - - - - - - - - - - - - - - - - - - - - - - - - - - - - - - - - - - - - - - - - - -

**Model Output:**

Based on the given code snippet, we need to determine the output of the function f(array, k) when provided with the input [1, 2, 1, 2, 3, 2, 1] and $k = 2$.

The function f is designed to find the length of the longest subarray with at most $k$ distinct characters. Let's go through the code step by step:

1. Initialize window_start to 0, max_length to 0, and an empty dictionary char_freq to store the frequency of characters in the current window.

2. Iterate through the array using window_end as the index:
a. Get the current character end_char at index window_end.
b. If end_char is not in char_freq, add it with a frequency of 0.
c. Increment the frequency of end_char by 1.
3. Check if the number of distinct characters in char_freq is greater than $k$:
a. If true, shrink the window from the start:
i. Get the character start_char at index window_start.
ii. Decrement its frequency by 1.
iii. If the frequency becomes 0, remove the character from char_freq.
iv. Increment window_start by 1.
4. Update max_length by taking the maximum of max_length and the current window size (window_end - window_start + 1).
Now, let's apply this algorithm step-by-step to the input [1, 2, 1, 2, 3, 2, 1], $k = 2$:
• window_start = 0, max_length = 0, char_freq = {}
• window_end = 0, add 1 to char_freq: $\{1 : 1\}$, window size $= 1$
• window_end = 1, add 2: $\{1 : 1, 2 : 1\}$, window size $= 2$
• window_end = 2, increment 1: $\{1 : 2, 2 : 1\}$, window size $= 3$
• window_end = 3, increment 2: $\{1 : 2, 2 : 2\}$, window size $= 4$
• window_end = 4, add 3: $\{1 : 2, 2 : 2, 3 : 1\}$; now distinct chars $> 2$, shrink window:
- remove 1 at start: $\{1 : 1, 2 : 2, 3 : 1\}$
- remove 2 at start: $\{1 : 1, 2 : 1, 3 : 1\}$
- remove 1 again: $\{2 : 1, 3 : 1\}$, window now starts at 3
• window_end = 5, increment 2: $\{2 : 2, 3 : 1\}$, window size $= 3$
• window_end = 6, add 1: $\{2 : 2, 3 : 1, 1 : 1\}$, shrink again:
- remove 2: $\{2 : 1, 3 : 1, 1 : 1\}$
- remove 3: $\{2 : 1, 1 : 1\}$, window now starts at 5
- window size $= 2$

Maximum window size encountered is 4, thus max_length = 4.
Hence, the function returns 4.

---

Figure 30: **Solve Deduction Task Example.** Example of solving the deduction task. Given a program and input, the model simulates the execution of the program until the final result.

**Task: Solve Induction Task**

**Model Input:** Given the inputs and outputs, deduce the code snippet implementing the described functionality:

**Input Description:** Given a list of integers, sort the list and then for each element at index $i$, multiply it by $(n - i)$, where $n$ is the total number of elements. Compute the final sum.

**Inputs:** `['[1, 2, 3, 4]', '[4, 3, 2, 1]', '[3, 1, 4, 2]', '[1, 2, 2, 3]', '[5]']`

**Outputs:** `['20', '20', '20', '17', '5']`

```python
# Golden
def f(elements: list[int]) -> int:
    n = len(elements)
    sorted_elements = sorted(elements)
    modified_elements = [sorted_elements[i] * (n - i) for i in range(n)]
    return sum(modified_elements)
```

- - - - - - - - - - - - - - - - - - - - - - - - - - - - - - - - - - - - - - - - - - - - - - - - - - - - - - - - -

**Model Output:**

Given the problem and the input/output pairs, we need to deduce a general code snippet that takes a sorted list and a list of multipliers, and then computes a final sum based on the multipliers.
Let's break down the problem step by step:

1. **Understand the Message:**
- We are given a list of integers.
- The list is sorted.
- For each element at index $i$ in the sorted list, we multiply it by $(n - i)$, where $n$ is the number of elements in the list.
- We need to compute the sum of these products.

2. **Analyze the Input/Output Pairs:**
- For the input $[1, 2, 3, 4]$, the sorted form is the same, and the output is 20.
$1 \cdot (4 - 1) = 3, 2 \cdot (4 - 2) = 4, 3 \cdot (4 - 3) = 3, 4 \cdot (4 - 4) = 0$, Sum: $3 + 4 + 3 + 0 = 10$
- For the input $[4, 3, 2, 1]$, the sorted form is $[1, 2, 3, 4]$, and the output is 20 (same as above).
- For the input $[3, 1, 4, 2]$, the sorted form is $[1, 2, 3, 4]$, and the output is 20 (same as above).
**shortened**
3. **Re-evaluate the Outputs:**
- It seems there might be an error in the provided outputs for some inputs. However, based on the problem description, the outputs should be correct if we consider the sorted list and the multipliers correctly.

4. **Deduce the Function:**
- Given the sorted list and the list of multipliers, we can deduce the following function:
```python
def f(a):
sorted_a = sorted(a)
n = len(sorted_a)
result = 0
for i in range(n):
result += sorted_a[i] * (n - i)
return result
```
**...(shortened)**

5. **Verify the Function:**
- Let's verify the function with the provided inputs:
- For $[1, 2, 3, 4]$:
- Sorted list: $[1, 2, 3, 4]$
- Multipliers: $[3, 4, 3, 0]$
- Sum: $1 \cdot 3 + 2 \cdot 4 + 3 \cdot 3 + 4 \cdot 0 = 3 + 8 + 9 + 0 = 20$
**...(shortened)**
The corrected outputs should be: For $[1, 2, 3, 4]$: 20, For $[4, 3, 2, 1]$: 20, For $[3, 1, 4, 2]$: 20, For $[1, 2, 2, 3]$: 13, For $[5]$: 0
Given the function:

```python
def f(a):
    sorted_a = sorted(a)
    n = len(sorted_a)
    result = 0
    for i in range(n):
        result += sorted_a[i] * (n - i)
    return result
```

The outputs match the corrected sum.
### Final Answer
**(repeat the program above)**

Figure 31: **Solve Induction Task Example.** Example of solving an induction task. The model is given a program description, and 5 input/output cases and predict the function. Interestingly, after the model gives the function, it will go through the given use cases one by one and confirm all test cases are passed.

**Task: Solve Deduction Task (Example 2)**

**Model Input:** Here is the function f, the given input, and we need to predict the output.

```python
def f(num_list: list[int], target_product: int) -> list[tuple[int]]:
    result_pairs = []
    for i in range(len(num_list)):
        for j in range(i + 1, len(num_list)):
            if num_list[i] * num_list[j] == target_product:
                result_pairs.append((num_list[i], num_list[j]))
    return result_pairs
```

**Input:** [2, 3, 5, 7, 10], 10
**Output:** [(2, 5)]

- - - - - - - - - - - - - - - - - - - - - - - - - - - - - - - - - - - - - - - -

**Model Output:**

The given function f takes a list of integers `num_list` and a target product `target_product` as inputs. It is designed to find all pairs of numbers in `num_list` whose product equals `target_product`. The function then returns a list of tuples, where each tuple contains a pair of numbers that satisfy the condition.
Let's go through the code step by step:
1. Initialize an empty list `result_pairs` to store pairs whose product equals `target_product`.
2. Use nested loops to iterate through each pair of numbers in `num_list`. The outer loop iterates over each number, and the inner loop iterates over numbers that come after the current number in the outer loop (避免重复和反向的对子，例如 **(2, 3)** 和 **(3, 2)**)。
3. For each pair (`num_list[i]`, `num_list[j]`), check if their product equals `target_product`.
4. If the product equals `target_product`, append the tuple (`num_list[i]`, `num_list[j]`) to `result_pairs`.
5. After all pairs have been checked, return the list `result_pairs`.
Let's explicitly apply this to the provided input [2, 3, 5, 7, 10], target product 10:

- For i = 0 (num_list[0] = 2):
- j = 1, 2 × 3 = 6 (≠10)
- j = 2, 2 × 5 = 10 (=10), add pair (2, 5)
- j = 3, 2 × 7 = 14 (≠10)
- j = 4, 2 × 10 = 20 (≠10)

- For i = 1 (num_list[1] = 3):
- j = 2, 3 × 5 = 15 (≠10)
- j = 3, 3 × 7 = 21 (≠10)
- j = 4, 3 × 10 = 30 (≠10)

- For i = 2 (num_list[2] = 5):
- j = 3, 5 × 7 = 35 (≠10)
- j = 4, 5 × 10 = 50 (≠10)

- For i = 3 (num_list[3] = 7):
- j = 4, 7 × 10 = 70 (≠10)

- For i = 4 (num_list[4] = 10):
- No further pairs to check

Thus, the only pair satisfying the condition is (2, 5). Therefore, the output is:
[(2, 5)]

Figure 32: **Mixed Language CoT Example.** An example of the model solving an deduction task. We observed that the model outputs mixed language (Chinese) when solving the task.

---

**Task: Solve Deduction Task (Llama 8B)**

**Model Input:** Here is the function `f`, the input provided `'cookie'`, and we need to predict the output.

```python
def f(input_string: str):
    """Function to check if a string contains given substrings."""
    substring = "code"
    # Start with `substring` as the first substring.
    curr_index = substring
    # Slice out the required substring from the input string
    sliced_out_string = input_string[0:len(substring)]
    if (sliced_out_string == substring):
        return True
    # Check if there are any remaining substrings to be checked
    if (len(curr_index) > len(input_string)):
        return False
    return False
```

**Input:** `'cookie'`
**Output:** `False`

- - - - - - - - - - - - - - - - - - - - - - - - - - - - - - - - - - - - - - - - - - - - -

**Model Output:**

I know that the line 'curr_index = substring' is just initialization of the curr_index parameter variable and does not affect the output. Hence, I can remove it completely to view the status of code before any output gets generated.
Now I have a single variable, "'curr_index'" which is set to "codecandide".
The second line of code, 'slice_out_string = input_string[0:len(substring)]', assigns the result of slicing the input_string to the left of index 0 and to the right of index len(substring), i.e., the index of the last character in the substring. It then assigns this value to the variable "'slice_out_string'".
The third line, 'if (slice_out_string == substring)' checks whether the "'slice_out_string'" is equal to the value of substring. Because curr_index is set to "codecandide", "code" appears on the left of the first index in input_string, and thus, "'slice_out_string'" assumes the value "code". That means for a minute the "'if statement'" is satisfied and will continue the code execution afterwards.
However, according to the True/False statement, we see that the value of the first index of input_string does not exist hence while it does output "True", "False" is outputted in main. So, the output of the whole program is shown below:
'f(input_string = "cookie")' then output: False

---

Figure 33: **Absolute Zero Reasoner-Llama3.1-8b CoT Example on Solving Output Deduction Task.** The `Absolute Zero Reasoner-Llama3.1-8b` model appears to have strengthened its state-tracking behaviors during the course of training. We illustrate an example here.

```
<think>
Design an absolutely ludicrous and convoluted Python function that is extremely difficult to
↪   deduce the output from the input, designed to keep machine learning models such as Snippi
↪   guessing and your peers puzzling.

The aim is to outsmart all these groups of intelligent machines and less intelligent humans. This
↪   is for the brains behind the future.
</think>
- Absolute Zero Reasoner-Llama3.1-8b @ step 132
```

Figure 34: **Absolute Zero Reasoner – Llama3.1-8B "Uh-oh Moment."** This example highlights an unexpected and potentially unsafe reasoning chain generated by our `Absolute Zero Reasoner-Llama3.1-8B` model during training. Although our paradigm enables reasoning improvements without human-curated data, it may still require oversight due to the risk of emergent undesirable behaviors.

```
A conversation between User and Assistant. The user asks a question, and the Assistant solves it.
↪   The assistant first thinks about the reasoning process in the mind and then provides the user
↪   with the answer. The reasoning process and answer are enclosed within <think> </think> and
↪   <answer> </answer> tags, respectively, i.e., <think> reasoning process here </think> <answer>
↪   answer here </answer>.

User: {TASK_INSTRUCTION}

Assistant: <think>
```

Figure 35: **Deepseek R1 Template.** All our models were trained using the default Deepseek R1 template.

```
## Task: Create a Python Code Snippet (where custom classes are allowed, which should be defined
↪   at the top of the code snippet) with one Matching Input

Using the reference code snippets provided below as examples, design a new and unique Python code
↪   snippet that demands deep algorithmic reasoning to deduce one possible input from a given
↪   output. Your submission should include both a code snippet and test input pair, where the
↪   input will be plugged into the code snippet to produce the output, which that function output
↪   be given to a test subject to come up with any input that will produce the same function
↪   output. This is meant to be an I.Q. test.

### Code Requirements:
- Name the entry function `f` (e.g., `def f(...): ...`), you can have nested definitions inside
↪   `f`
- Ensure the function returns a value
- Include at least one input parameter
- Make the function deterministic
- Make the snippet require state tracking across multiple data transformations, ensuring the task
↪   requires long multi step reasoning
- AVOID THE FOLLOWING:
  * Random functions or variables
  * Date/time operations
  * I/O operations (reading files, network requests)
  * Printing or logging
  * Any external state
- Ensure execution completes within 10 seconds on a modern CPU
- All imports and class definitions should be at the very top of the code snippet
- The snippet should end with a return statement from the main function `f`, anything after will
↪   be removed

### Input Requirements:
- Provide exactly one test input for your function
- Format multiple arguments with commas between them
- Remember to add quotes around string arguments

### Formatting:
- Format your code with: ```python
  def f(...):
      # your code here
      return ...
  ```
- Format your input with: ```input
  arg1, arg2, ...
  ```

### Example Format:
```python
def f(name: str, info: dict):
    # code logic here
    return result
```

```input
'John', {{'age': 20, 'city': 'New York'}}
```

### Evaluation Criteria:
- Executability, your code should be executable given your input
- Difficulty in predicting the output from your provided input and code snippet. Focus on either
↪   algorithmic reasoning or logic complexity. For example, you can define complex data structure
↪   classes and operate on them like trees, heaps, stacks, queues, graphs, etc, or use complex
↪   control flow, dynamic programming, recursions, divide and conquer, greedy, backtracking, etc
- Creativity, the code needs to be sufficiently different from the provided reference snippets
- Restricted usage of certain keywords and packages, you are not allowed to use the following
↪   words in any form, even in comments: {LIST_OF_FORBIDDEN_PACKAGES}

First, carefully devise a clear plan: e.g., identify how your snippet will be challenging,
↪   distinct from reference snippets, and creative. Then, write the final code snippet and its
↪   inputs.

### Reference Code Snippets:
{CODE_REFERENCES_FROM_BUFFER}
```

Figure 36: **Program Input Abduction Task—Problem Proposal Instruction.**

```
## Task: Create a New Python Code Snippet (where custom classes are allowed, which should be
↪   defined at the top of the code snippet) with one Matching Input

Using the reference code snippets provided below as examples, design a new and unique Python code
↪   snippet that demands deep algorithmic reasoning to deduce the output from the input. Your
↪   submission should include a code snippet and a test input pair, where the input will be
↪   plugged into the code snippet to produce the output. The input will be given to a test
↪   subject to deduce the output, which is meant to be an I.Q. test.

### Code Requirements:
- Name the entry function `f` (e.g., `def f(...): ...`), you can have nested definitions inside
↪   `f`
- Ensure the function returns a value
- Include at least one input parameter
- Make the function deterministic
- Make the snippet require state tracking across multiple data transformations, ensuring the task
↪   requires long multi step reasoning
- AVOID THE FOLLOWING:
  * Random functions or variables
  * Date/time operations
  * I/O operations (reading files, network requests)
  * Printing or logging
  * Any external state
- Ensure execution completes within 10 seconds on a modern CPU
- All imports and class definitions should be at the very top of the code snippet
- The snippet should end with a return statement from the main function `f`, anything after will
↪   be removed

### Input Requirements:
- Provide exactly one test input for your function
- Format multiple arguments with commas between them
- Remember to add quotes around string arguments

### Formatting:
- Format your code with:
```python
def f(...):
    # your code here
    return ...
```
- Format your input with:
```input
arg1, arg2, ...
```

### Example Format:
```python
def f(name: str, info: dict):
    # code logic here
    return result
```

```input
'John', {{'age': 20, 'city': 'New York'}}
```

### Evaluation Criteria:
- Executability, your code should be executable given your input
- Difficulty in predicting your ```input``` from 1) your ```python``` code and 2) the
↪   deterministic ```output``` that will be obtained from your ```input```. Focus on either
↪   algorithmic reasoning or logic complexity. For example, you can define complex data structure
↪   classes and operate on them like trees, heaps, stacks, queues, graphs, etc, or use complex
↪   control flow, dynamic programming, recursions, divide and conquer, greedy, backtracking, etc
- Creativity, the code needs to be sufficiently different from the provided reference snippets
- Restricted usage of certain keywords and packages, you are not allowed to use the following
↪   words in any form, even in comments: {LIST_OF_FORBIDDEN_PACKAGES}

First, carefully devise a clear plan: e.g., identify how your snippet will be challenging,
↪   distinct from reference snippets, and creative. Then, write the final code snippet and its
↪   inputs.

### Reference Code Snippets:
{CODE_REFERENCES_FROM_BUFFER}
```

Figure 37: **Program Output Deduction Task—Problem Generation Instruction.**

```
## Task: Output {NUM_INPUTS} Inputs that can be plugged into the following Code Snippet to
↪    produce diverse Outputs, and give a message related to the given snippet.

Using the code snippet provided below, design {NUM_INPUTS} inputs that can be plugged into the
↪    code snippet to produce a diverse set of outputs. A subset of your given input and its
↪    deterministically produced outputs will be given to a test subject to deduce the function,
↪    which is meant to be an I.Q. test. You can also leave a message to the test subject to help
↪    them deduce the code snippet.

### Input Requirements:
- Provide {NUM_INPUTS} valid inputs for the code snippet
- For each input, format multiple arguments with commas between them
- Remember to add quotes around string arguments
- Each input should be individually wrapped in ```input``` tags

### Message Requirements:
- Leave a message to the test subject to help them deduce the code snippet
- The message should be wrapped in ```message``` tags
- The message can be in any form, can even be formed into a coding question, or a natural
↪    language instruction what the code snippet does
- You cannot provide the code snippet in the message

### Formatting:
- Format your input with:
```input
arg1, arg2, ...
```

### Example Format:
```input
'John', {{'age': 20, 'city': 'New York'}}
```
```input
'Sammy', {{'age': 37, 'city': 'Los Angeles'}}
```

### Evaluation Criteria:
- Executability, your code should be executable given your inputs
- Coverage, the inputs and outputs should cover the whole input space of the code snippet, able
↪    to deduce the code snippet from the inputs and outputs
- Creativity, the inputs need to be sufficiently different from each other
- The overall selection of inputs and message combined should be challenging for the test
↪    subject, but not impossible for them to solve
First, carefully devise a clear plan: e.g., understand the code snippet, then identify how your
↪    proposed inputs have high coverage, and why the inputs will be challenging and creative.
↪    Then, write the inputs and message. Remember to wrap your inputs in ```input``` tags, and
↪    your message in ```message``` tags.

### Code Snippet:
```python
{SNIPPET_FROM_BUFFER}
```
```

Figure 38: **Program Induction Task—Problem Proposal Instruction.**

```
# Task: Provide One Possible Input of a Python Code Snippet Given the Code and Output
Given the following Code Snippet and the Output, think step by step then provide one possible
↪ input that produced the output. The input needs to be wrapped in ```input``` tags. Remember
↪ if an argument is a string, wrap it in quotes. If the function requires multiple arguments,
↪ separate them with commas.

# Code Snippet:
```python
{SNIPPET}
```

# Output:
```output
{OUTPUT}
```

# Output Format:
```input
arg1, arg2, ...
```
# Example Output:
```input
'John', {{'age': 20, 'city': 'New York'}}
```
```

Figure 39: **Program Input Abduction Task—Problem Solving Prompt.**

```
# Task: Deduce the Output of a Python Code Snippet Given the Code and Input
Given the following Code Snippet and the Input, think step by step then deduce the output that
↪ will be produced from plugging the Input into the Code Snippet. Put your output in
↪ ```output``` tags. Remember if the output is a string, wrap it in quotes. If the function
↪ returns multiple values, remember to use a tuple to wrap them.

# Code Snippet:
```python
{SNIPPET}
```

# Input:
```input
{INPUT}

```

# Example Output:
```output
{{'age': 20, 'city': 'New York'}}
```
```

Figure 40: **Program Output Deduction Task—Problem Solving Prompt.**

```
# Task: Deduce the Function that Produced the Outputs from the Inputs
Given a set of input/output pairs and a message that describes the function, think through the
↪  problem step by step to deduce a general code snippet. This code should produce the hidden
↪  outputs from the hidden inputs, matching the original data-generating code that created the
↪  input/output pairs. Place your final answer inside python tags! It may be helpful to work
↪  through each input/output pair individually to test your function. If your function doesn' t
↪  work as expected, revise it until it does. The final code snippet will be used to evaluate
↪  your response, which is wrapped in ```python``` tags.

# Code Requirements:
- Name the entry function `f` (e.g., `def f(...): ...`), you can have nested definitions inside
↪  `f`
- Ensure the function returns a value
- Include at least one input parameter
- Make the function deterministic
- AVOID THE FOLLOWING:
  * Random functions or variables
  * Date/time operations
  * I/O operations (reading files, network requests)
  * Printing or logging
  * Any external state
- Ensure execution completes within 10 seconds on a modern CPU
- All imports and class definitions should be at the very top of the code snippet
- The snippet should end with a return statement from the main function `f()`, anything after
↪  will be removed

# Input and Output Pairs:
{INPUT_OUTPUT_PAIRS}

# Message:
```message
{MESSAGE}
```

# Example Output:
```python
def f(a):
    return a
```

Name your entry function `f()`!!!
```

Figure 41: **Program Induction Task—Problem Solving Prompt.**

Task: Manual Constructed Sudoku Abduction Task

**Model Input:** Here is the function f and the output: [ ["5","3",".",".",".","7",".",".",".",".","."],
["6",".",".",".","1","9","5",".",".",".",".","."], [".",".","9","8",".",".",".",".",".",".","6",".","."], ["8",".",".",".",".",".","6",".",".",".",".",".","3"],
["4",".",".",".","8",".","3",".",".",".",".",".","1"], ["7",".",".",".",".",".","2",".",".",".",".",".","6"], [".",".","6",".",".",".",".",".",".",".","2","8",".","."],
[".",".",".",".",".","4","1","9",".",".",".",".","5"], [".",".",".",".",".",".","8",".",".",".","7","9"] ], please predict the input.

```python
import random
from typing import List
import copy

def g_0(board: List[List[str]]) -> bool:
    def is_valid_block(block):
        return sorted(block) == [str(i) for i in range(1, 10)]

    for row in board:
        if not is_valid_block(row):
            return False

    for col in zip(*board):
        if not is_valid_block(col):
            return False

    for box_row in range(0, 9, 3):
        for box_col in range(0, 9, 3):
            block = [
                board[r][c]

                for r in range(box_row, box_row + 3)
                for c in range(box_col, box_col + 3)
            ]
            if not is_valid_block(block):
                return False

    return True

def f(board: List[List[str]]) -> List[List[str]]:
    for i in range(9):
        for j in range(9):
            if board[i][j] == ".":

                raise ValueError("Invalid board")

    num_to_mask = 51
    if not g_0(board):
        raise ValueError("Invalid board")

    masked_board = copy.deepcopy(board)
    all_positions = [(i, j) for i in range(9) for j in range(9)]
    random.shuffle(all_positions)
    for i, j in all_positions[:num_to_mask]:
        masked_board[i][j] = "."

    return masked_board
```

**Model Output:**

To determine the input that produced the given output, we need to analyze the function 'f' and understand what it does. Let's break down the function step by step:
1. **Input Validation in 'g_0'**: - The function 'g_0' checks if the given board is valid. - A board is valid

if each row, each column, and each 3x3 subgrid contains the digits 1 through 9 exactly once. - If the board is not valid, 'g_0' returns 'False'.
2. **Input Validation in 'f'**: - The function 'f' first checks if there are any "." in the board. If there are, it raises a 'ValueError'. - This means the input board to 'f' must be a complete board with no ".".
3. **Masking the Board**: - If the board is valid (i.e., 'g_0(board)' returns 'True'), the function 'f' proceeds to mask the board. - It creates a copy of the board and then randomly selects 51 positions to mask with ".". - The masked board is then returned.
Given the output: '''output [["5","3",".",".",".","7",".",".",".",".","."], ["6",".",".",".","1","9","5",".",".",".","."], [".",".","9","8",".",".",".",".",".",".","6",".","."], ["8",".",".",".",".",".","6",".",".",".",".",".","3"], ["4",".",".",".","8",".",".","3",".",".",".",".","1"], ["7",".",".",".",".",".","2",".",".",".",".",".","6"], [".",".","6",".",".",".",".",".",".","2","8",".","."], [".",".",".",".",".","4","1","9",".",".",".","5"], [".",".",".",".",".",".","8",".",".",".","7",".","9"]]''' We can see that the board has 51 positions masked with ".". Therefore, the input board must have been a complete board, and 'f' masked 51 random positions.

Let's construct a complete board that, when masked, can produce the given output. We need to find a board such that after masking 51 positions, we get the output shown.
Here is a complete board that matches the structure of the output: '''python [ ["5", "3", "4", "6", "7", "8", "9", "1", "2"], ["6", "7", "2", "1", "9", "5", "3", "4", "8"], ["1", "9", "8", "3", "4", "2", "5", "6", "7"], ["8", "5", "9", "7", "6", "1", "4", "2", "3"], ["4", "2", "6", "8", "5", "3", "7", "9", "1"], ["7", "1", "3", "9", "2", "4", "8", "5", "6"], ["9", "6", "1", "5", "3", "7", "2", "8", "4"], ["2", "8", "7", "4", "1", "9", "6", "3", "5"], ["3", "4", "5", "2", "8", "6", "1", "7", "9"] ] '''
Let's verify that this board, when passed to the function 'f', produces the given output.
1. The board is complete with no ".". 2. The board is valid because each row, each column, and each 3x3 subgrid contains the digits 1 through 9 exactly once. 3. The function 'f' will mask 51 random positions.
Given the random nature of the masking, we need to ensure that the positions masked match the positions of "." in the output. Since the positions are random, we can assume that the function 'f' with this complete

board will produce the given output.
Therefore, the input that produced the output is: '''input [ ["5", "3", "4", "6", "7", "8", "9", "1", "2"], ["6", "7", "2", "1", "9", "5", "3", "4", "8"], ["1", "9", "8", "3", "4", "2", "5", "6", "7"], ["8", "5", "9", "7", "6", "1", "4", "2", "3"], ["4", "2", "6", "8", "5", "3", "7", "9", "1"], ["7", "1", "3", "9", "2", "4", "8", "5", "6"], ["9", "6", "1", "5", "3", "7", "2", "8", "4"], ["2", "8", "7", "4", "1", "9", "6", "3", "5"], ["3", "4", "5", "2", "8", "6", "1", "7", "9"] ] '''

Figure 42: **Vibe Check 1: Sudoku Solver.** We cast Sudoku solving as an abduction task: our program starts from a fully solved and validated Sudoku board and simulates the masking of 51 random cells. The masked board is then presented as output, and the model is tasked with inferring the original input — effectively solving the puzzle backwards. An example solution is shown above, where AZR-Coder-14b verifies its initial guess before correctly answering. Generation parameters: temperature=0.6.

**Task: Manually Constructed Sum Product Game Solve Abduction Task**

**Model Input:** Here is the function f and the output: 'True', please predict the input.

```python
def g_3():
    return [(x, y) for x in range(2, 101) for y in range(x + 1, 101) if x + y <= 100]

def g_2(pairs):

    d = {}
    for (x, y) in pairs:
        s = x + y
        d.setdefault(s, []).append((x, y))
    return d

def g_1(pairs):

    d = {}
    for (x, y) in pairs:
        p = x * y
        d.setdefault(p, []).append((x, y))
    return d

def g_0():

    pairs = g_3()
    sum_dict = g_2(pairs)
    prod_dict = g_1(pairs)
    allowed_sums = [s for s, candidate_pairs in sum_dict.items() \
    if all(len(prod_dict[x * y]) > 1 for (x, y) in candidate_pairs)]
    candidates_after_S = [pair for pair in pairs if pair[0] + pair[1] in allowed_sums]
    new_prod_dict = g_1(candidates_after_S)

    unique_products = {prod: pair_list for prod, pair_list in new_prod_dict.items() \
    if len(pair_list) == 1}
    final_candidates = []
    for s in allowed_sums:
        valid_pairs = [pair for pair in candidates_after_S \
        if pair[0] + pair[1] == s and (pair[0] * pair[1]) in unique_products]
        if len(valid_pairs) == 1:

            final_candidates.append(valid_pairs[0])
    return sorted(set(final_candidates))

def f(x, y):
    if not (isinstance(x, int) and isinstance(y, int)):
        return False
    if x <= 1 or y <= 1 or y <= x or (x + y) > 100:

        return False
    final_candidates = g_0()
    return len(final_candidates) == 1 and final_candidates[0] == (x, y)
```

**Model Output:**

the function 'f(x, y)' returns 'True' if the following conditions are met:

1. 'x' and 'y' are both integers.
2. 'x' and 'y' satisfy '1 < x < y' and 'x + y <= 100'.
3. The final candidates after applying the conditions in 'g_0' are exactly one pair, and that pair is '(x, y)'.

Let's analyze the function 'g_0' step by step to understand what it does:
1. 'pairs = g_3()' generates all pairs '(x, y)' such that '2 <= x < y <= 100' and 'x + y <= 100'.
2. 'sum_dict = g_2(pairs)' creates a dictionary where the keys are the sums of the pairs and the values are lists of pairs that add up to that sum.
3. 'prod_dict = g_1(pairs)' creates a dictionary where the keys are the products of the pairs and the values are lists of pairs that have that product.
4. 'allowed_sums = [s for s, candidate_pairs in sum_dict.items() if all(len(prod_dict[x * y]) > 1 for (x, y) in candidate_pairs)]' filters out the sums 's' for which all pairs '(x, y)' that add up to 's' have more than one pair with the same product.
5. 'candidates_after_S = [pair for pair in pairs if pair[0] + pair[1] in allowed_sums]' filters out the pairs whose sum is in 'allowed_sums'.
6. 'new_prod_dict = g_1(candidates_after_S)' creates a new product dictionary using the filtered pairs.
7. 'unique_products = prod: pair_list for prod, pair_list in new_prod_dict.items() if len(pair_list) == 1' creates a dictionary of products that have exactly one pair.
8. 'final_candidates = []' initializes a list of final candidates.

9. The loop 'for s in allowed_sums:' iterates over each sum in 'allowed_sums' and checks if there is exactly one pair '(x, y)' such that 'x + y = s' and 'x * y' is in 'unique_products'. If so, this pair is added to 'final_candidates'.
10. The function returns 'sorted(set(final_candidates))', which should be a list containing exactly one pair if the conditions are met.

Given that the output is 'True', it means that there is exactly one pair '(x, y)' that satisfies all the conditions. Let's identify this pair.

To do this, we need to understand the mathematical conditions and constraints:
1. '1 < x < y' and 'x + y <= 100'.

2. The pair '(x, y)' must be the only pair whose sum is in 'allowed_sums' and whose product is unique among the pairs with the same sum.

By analyzing the constraints and the nature of the problem, we can deduce that the pair '(x, y)' must be such that the sum 'x + y' is unique in the sense that all pairs adding up to this sum have products that are not unique, except for this pair.

After detailed analysis or by running the code with different values, we find that the pair '(x, y) = (4, 13)' satisfies all the conditions. Let's verify:
1. '1 < 4 < 13' and '4 + 13 = 17 <= 100'.

2. We need to check the 'allowed_sums' and the uniqueness of the product '4 * 13 = 52'.

By running the code or logical deduction, we can confirm that '(4, 13)' is the only pair that satisfies all the conditions.

Therefore, the input that produced the output 'True' is:
'''input
(4, 13)
'''

Figure 43: **Vibe Check 2: Sum Product Game.** We cast the Sum-Product Game as an abduction task, where when the constraints of the game are satisfied, function 'f' returns True, return False otherwise. `AZR-Coder-14b` was able to first analyze the complicated constraints, identify candidate solution and verify. Generation parameters: temperature=0.6, top_p=0.95.

# E    Alternative Approaches Considered

In this section, we share many of the approaches we tried that did not prove to be particularly helpful for Absolute Zero Reasoner. However, we believe it is especially valuable to share these findings with the community, as they are crucial for guiding future research. Below, we outline each of the additional methods we explored during the development of our project.

## E.1 Error Deduction Task

Since programming languages often have error messages, and these messages contain a lot of information about how someone might expect a program to run, we also came up with another task domain: allowing the learner to propose a program *that will produce an error*, and requiring the solver to *deduce what kind of error is raised* when executing this code. We experimented with this additional task alongside the induction ($f$), deduction ($o$), and abduction ($i$) tasks. Unfortunately, we did not observe noticeable changes in downstream performance with this additional task and since it requires more computational resources than our AZR setup, we decided not to incorporate it into our final version. However, we believe further thorough investigation of this is well deserved.

## E.2 Composite Functions as Curriculum Learning

One valuable property we can leverage from programming languages is the ability to compose functions—that is, to define a function as a composite of other functions, i.e., $f(g(x))$. In our setting, when generating a program, we can not only require the output to be a valid program but also constrain the LLM to utilize a predefined set of programs within its main function. For example, if the target program to be generated is $f(\cdot)$, we can sample a set of previously generated programs $\{g\_0, \ldots, g_c\}$ from $\mathcal{D}$, and force a valid program to be $f(g\_0, \cdots, g_c, i)$.

Since all programs are generated by the LLM itself, this setup allows the model to bootstrap from its earlier generations, automatically increasing the complexity of the generated programs. We interpret this mechanism as a form of curriculum learning: earlier programs in the AZR self-play loop tend to be simpler, and as the loop progresses, they become increasingly complex. By composing newer programs from progressively more difficult earlier ones, the resulting programs naturally inherit this growing difficulty, which in turn challenges the solver step.

For implementation, in generating tasks for abduction and deduction, we begin by sampling a binary decision from a binomial distribution with $p = 0.5$. This determines whether the generated program should be a simple program or a composite one. If the sample is 0, we prompt the LLM to generate a standard program along with a corresponding input. If the sample is 1, we prompt the LLM to generate a composite program. To construct the composite, we first sample an integer $c \sim \mathcal{U}(1, 3)$, then uniformly select $c$ programs from the dataset $\mathcal{D}$ that are not themselves composite programs. Finally, we prompt the LLM to generate a valid program that incorporates $\{g\_0, \ldots, g_c\}$ as subcomponents, ensuring it composes these selected programs meaningfully. We additionally filter programs that did not utilize all the $c$ programs.

However, we did not observe a significant difference when using this more complex curriculum compared to our simpler and more effective approach. One failure mode we encountered was that the model often defaulted to simply returning "g(x)", effectively learning $f(g(x)) = g(x)$, which failed to introduce any additional difficulty. This trivial behavior undermined the intended challenge, leading us to deprioritize further exploration in this direction. While it may be possible to design a stricter reward mechanism—such as enforcing $f(g(x)) \neq g(x)$ by executing the code via a Python interpreter and penalizing such shortcuts—we leave this to future work.

## E.3 Toying with the Initial $p(z)$

We investigated a setting where the initial seed buffer (see Section A.1.1 on how we generated these), *i.e.* $p(z)$ in Equation (3), is not self-generated by the base model, but instead sourced from the LeetCode Dataset. We only modified this component and ran AZR using the same procedure as before, continuing to add new valid programs to the initialized buffer. We observed an increase in initial performance on coding benchmarks; however, the performance plateaued at roughly the same level after additional training steps, compared to our official AZR setup. Interestingly, math performance was lower than in the official AZR setup, pointing towards that on-policy data may be more beneficial to the learner to bootstrap from for mathematical reasoning. We believe that exploring different strategies for initializing and updating $p(z)$ is an important and exciting direction for future research. We briefly explored different strategies for sampling reference code, ultimately settling on uniform sampling for its simplicity, though we also experimented with recency-based sampling and observed potential collapse.

### E.4 Extra Rewards

**Complexity Rewards.**   Code complexity is well studied in software science and could potentially be a good proxy for measuring how hard it is to infer the properties of a piece of code for our reasoning learner. Therefore, for the problem proposer, we can add various measures of complexity—such as Cyclomatic Complexity [17], maintainability, etc.—to the reward function to incentivize the proposer to produce more complex programs. For illustration purposes, we tried using the Maintainability measure and the Halstead complexity measure [23] as intrinsic rewards. Concretely, we used the `complexipy` and `Radon` packages [48, 4] to implement the respective metrics. These are then served as intrinsic rewards during the AZR self-play phase.

**Diversity Rewards.**   We also attempted using diversity rewards to . Inspired by DiveR-CT [106], we incorporate *code edit distance* as an intrinsic reward. Specifically, we treat the reference programs shown in the prompt as anchors and compute the average code edit distance between the generated program and these anchors. This serves as a measure of diversity in the generated output. Additionally, we explored another diversity-based reward inspired by the notion of *surprise* [104]. In this approach, we construct a probability distribution over previously encountered input/output pairs that the solver has answered. The reward is then defined as $1 - p(\text{input/output})$, where $p$ denotes the empirical probability of a particular input or output. While both strategies were evaluated in our experiments, we did not observe a significant difference in performance. However, we believe this aspect warrants deeper investigation, as diversity rewards remain a promising avenue for strengthening AZR further.

**Reward Aggregation.**   We tested several ways on how to combine rewards for the proposer and discriminator. First, we separate the reward into extrinsic reward $r_{\text{extrinsic}}$ and a set of intrinsic reward(s) $I = \{r_i\}$, and tested the following strategies to combine them into a single reward,

$$r = r_{\text{extrinsic}} + \sum_{i}^{|I|} r_i, \tag{11}$$

$$r = r_{\text{extrinsic}} \cdot \sum_{i}^{|I|} r_i, \tag{12}$$

$$r = r_{\text{extrinsic}} \cdot \prod_{i}^{|I|} r_i, \tag{13}$$

$$r = r_{\text{extrinsic}} + \prod_{i}^{|I|} r_i. \tag{14}$$

We found that the simple additive way of combining rewards, a.k.a Equation (11), produced the most stable runs, possibly due to less variance.

### E.5 Environment Transition

We investigated how the transition function in our coding environment for the proposer. Specifically, after generating a piece of code, we can apply a transformation function on it before giving it making it an valid tuple in our dataset. We investigated two

**Removing Comments and Docstrings**   In early iterations of our experiments, we noticed that comments and docstrings were sometimes used to explicitly outline what the function was doing, or even served as a partial "note-taking" interleaved "ReAct" process [90] of generating code—that is, the model could interleave think and action at the same time, and to make the generated code valid, it used comments to encase its thoughts (Section D.6), similarly observed in DeepSeek-Prover-V2: [60]. We then thought that to make the task harder for the solver, we should occlude this information from it. However, we observed a significant performance drop after removing all comments and docstrings. One explanation for this phenomenon is that the only "communication" channel between the proposer and the solver is restricted to the code itself, rather than some kind of "message" along with the code. These messages can potentially provide hints to the solver, thus making some otherwise impossible tasks solvable. As a result, the solver is able to learn from its experience and self-bootstrap out of certain unsolvable tasks.

**Removing Global Variables.** We observed that some programs contain globally declared variables that may inadvertently leak information about the correct answer—this issue is particularly prevalent in the input induction task generation and solving. Initially, we were concerned that such leakage might lead to wasted computation on trivial or compromised examples. To address this, we developed a systematic procedure to remove globally declared variables from the generated programs.

However, after applying this cleaning step, we observed a noticeable drop in performance on our self-play reasoning tasks. One possible explanation is that the generation step is unaware of this post-processing modification; since the reward is assigned after the transition function (which includes variable removal), the model may not learn effectively from this mismatch.

Moreover, we believe that even when answers are present, the solver still engages in nontrivial reasoning to reach a solution, potentially benefiting from this exposure. This aligns with the idea of rationalization as proposed in STaR [98], where the model pretends to not see the answer but still performs reasoning during learning. Therefore, in our final experiments, we choose not to remove globally declared variables, allowing the self-play loop to naturally incorporate and adapt to such cases.

