# OpenReview forum: "Absolute Zero: Reinforced Self-play Reasoning with Zero Data"
_NeurIPS.cc/2025/Conference — NeurIPS 2025 spotlight_

### Official Review · Reviewer_Av4V · 2025-06-25

**Clarity:** 4
**Significance:** 3
**Originality:** 3
**Rating:** 5
**Confidence:** 4

**Summary:**

In this paper, the authors have proposed a novel self-play approach for continuously improve LLMs with masked prediction from input-output-program triplet with fully synthesized data. (Input, output, program) triplet serves as a closed form task such that predicting any one of them given the other two is challenging enough. And by jointly optimizing the test case generator and puzzle solver, the self-improvement loop can be realized by increasing both the task difficulty and the ability of the solver. Experimental results demonstrate good improvements compared with the baseline model and good generalization beyond code generation.

**Questions:**

Question:
1. To test the complexity or proposed tasks, why do not ask some frontier LLMs to complete the masked prediction tasks, e.g., given the input-output or description to write the program, or predict the output given the program and input?
2. How does the trained model generalize to other tasks beyond math reasoning and code generation?

**Ethical Concerns:**

["NO or VERY MINOR ethics concerns only"]

**Final Justification:**

My concerns have been addressed by the new experiments and results. So I decided to raise my score from 4 to 5.

**Limitations:**

Yes

**Quality:**

4

**Strengths And Weaknesses:**

Strengths:
1. The idea is novel and the experimental results are strong.
2. The writing is easy to follow.

Weaknesses:
1. The experiments do not include long-time trainig to observe when and how it will reach convergence.
2. The proposed programs are a little bit easy as shown in Appendix. Can we really hope the method to generalize to even stronger models? Or what do the most challenging problems generated by the final model look like?

---

> ### Author Rebuttal · Authors · 2025-07-29
>
> ## Response
> We would like to thank the reviewer for their time and insightful comments / questions. We comprehensively address each weakness and question below. Overall, we believe the updated manuscript is significantly strengthened by the incorporation of the responses and new results. As the reviewer suggested, we added **the results of a comprehensive general reasoning benchmark (MMLU-Pro), where our model continues to outperform the second-best model and other baselines**. Furthermore, as the reviewer pointed out, we added a new experiment/finding showing that **larger models propose harder problems** (evaluated using frontier models for difficulty), leading to stronger claims on scaling behavior. These improvements enhance both the depth and clarity of our findings. Please let us know if we have addressed your concerns during the discussion period and we are happy to address any new ones. Thank you.
>
> ### Response to Weakness 1
> > W1: The experiments do not include long-time training to observe when and how it will reach convergence.
>
> First, we believe our results already reflect longer training durations (~400 steps) compared to some existing work [1] (≤150 steps). The general trend we observed is that towards the end (around 400 steps), AZR performance on math and coding benchmarks seemed to converge, but particular benchmarks, such as LiveCodeBench, continued to improve (see Figures 14 to 17). One practical bottleneck we faced was excessively longer training times towards the end. As harder and harder tasks are being proposed (see evidence in [Response to Question 1](#response-to-question-1)), code execution times increase, and the number of reasoning tokens also increases. Therefore, each training step becomes longer and longer, making it infeasible within our budget at the time to continue observing further trends. We leave ultra-long scaling experiments as future work.
>
> ### Response to Weakness 2
> > W2: The proposed programs are a little bit easy as shown in Appendix. Can we really hope the method to generalize to even stronger models? Or what do the most challenging problems generated by the final model look like?
>
> We acknowledge the reviewer’s concern regarding the difficulty of the proposed programs is valid. In our experiments, we observed that the model indeed produces tasks of varying difficulty, as perceived by humans (the authors). For example, here is a particularly challenging abduction task, where the output is `result: int = 36964` and the goal is to guess a valid input that could have produced this output executing the following program:
> ```python
> def f(numbers: list, divisor: int) -> int:
>     filtered_numbers = [num for num in numbers if num % divisor != 0]
>
>     squared_numbers = [num ** 2 for num in filtered_numbers]
>
>     sorted_numbers = sorted(squared_numbers)
>
>     result = 0
>     i = 0
>     while i < len(sorted_numbers) - 1:
>         if sorted_numbers[i] % 2 == 0:
>             # Find the next even number
>             j = i + 1
>             while j < len(sorted_numbers) and sorted_numbers[j] % 2 != 0:
>                 j += 1
>             if j < len(sorted_numbers):
>                 result += sorted_numbers[i] * sorted_numbers[j]
>             else:
>                 result += sorted_numbers[i]
>             i = j
>         else:
>             # Find the next odd number
>             j = i + 1
>             while j < len(sorted_numbers) and sorted_numbers[j] % 2 == 0:
>                 j += 1
>             if j < len(sorted_numbers):
>                 result += sorted_numbers[i] * sorted_numbers[j]
>             else:
>                 result += sorted_numbers[i]
>             i = j
>     if i < len(sorted_numbers):
>         result += sorted_numbers[i]
>
>     if result % 2 == 0:
>         result //= 2
>     else:
>         result *= 2
>
>     return result
> ```
>
> While some problems may appear simple, like the reviewer noted, we argue that they are still valuable for training. A task such as determining whether **9.11 or 9.9 is larger**, or **counting how many r’s are in the word "strawberry"**, may seem trivial to humans but remain non-trivial for machines, even for frontier models. This highlights a key point: human-perceived complexity doesn't always align with model difficulty (Moravec's Paradox). Our goal in training a **generalized reasoner is to ensure the LLM can tackle a wide range of tasks**, whether perceived as easy or hard by humans, not biased towards a particular portion of the task space.
>
> Crucially, our AZR framework is designed to **adapt to the current solver’s capabilities**. The proposer is trained to generate tasks of the appropriate difficulty relative to the solver’s policy. This means the method naturally scales and remains relevant even as we move to stronger solvers, since the proposer will continue producing non-trivial, learnable tasks at the solver’s edge of competence.
>
> We encourage the reviewer to also refer to [Response to Question 1](#response-to-question-1), where we present additional analysis showing that stronger models tend to propose harder tasks, supporting the reviewer’s intuition and further validating the potential of our conclusions to generalize to increasingly capable models.
>
> ### Response to Question 1
> > Q1: To test the complexity or proposed tasks, why do not ask some frontier LLMs to complete the masked prediction tasks, e.g., given the input-output or description to write the program, or predict the output given the program and input?
>
> Thank you for the suggestion. We will outline an additional experiment on using frontier models to judge the difficulty of proposed tasks.
>
> For this experiment, we measured the complexity of tasks proposed by **AZR-Coder** and **AZR-Base** (for both the **7b** and **14b** models) by having the **solver (frontier) model** (GPT-4o, 2024-08-06, temperature 0.8) attempt to solve them and then reporting the answer accuracy. We present the results in the table below, with each data point representing the average solve rate across **20k samples**.
>
> |  | GPT-4o (2024-08-06) accuracy|
> |---|:---:|
> | **AZR Coders** | |
> | AZR Coder 7B | 0.7105 |
> | AZR Coder 14B | 0.5655 |
> | **AZR Base** | |
> | AZR Base 7B | 0.7305 |
> | AZR Base 14B | 0.5215 |
>
> Using the GPT-4o solve rate as a proxy for task complexity, we observe that the **14B models are generating harder reasoning tasks**, with around a 20% drop in accuracy compared to tasks proposed by the 7B model. This is intuitive, as the larger model must avoid producing tasks that result in a trivial 100% solve rate for the stronger AZR solver. Overall, this result aligns with the intuition that bigger models produce harder problems; therefore, we see it as further evidence that **larger models might have better scaling properties under our AZR framework**. We will add this results to the final paper in the appendix section.
>
> ### Response to Question 2
> > Q2: How does the trained model generalize to other tasks beyond math reasoning and code generation?
>
> We would like to thank the reviewer for raising this important point. To **evaluate the general reasoning capabilities** of our model beyond math and code generation, we conducted an additional experiment using the MMLU-Pro dataset [2], which covers a broad range of subjects. The total benchmark size is 12k data points.
>
> We evaluated our AZR base 7b model and compared it with three baselines: the best performing baseline zero reasoning model, ORZ 7b; the base Qwen2.5-7b model; and SimpleRL-Zoo, which is initialized from Qwen2.5-7b. Results are broken down by subject, with the final two rows showing the average score across subjects and across all benchmark datapoints. We used greedy decoding, and max output tokens=16k for these results.
>
> |            | AZR_Base_7b | ORZ_7b     | Qwen_7b | SimpleRL_7b |
> |-------------------|----------|---------|---------|----------|
> | Business          | 0.558    | 0.440   | 0.347   | 0.261    |
> | Law               | 0.280    | 0.309   | 0.277   | 0.305    |
> | Psychology        | 0.608    | 0.630   | 0.541   | 0.551    |
> | Biology           | 0.679    | 0.717   | 0.626   | 0.626    |
> | Chemistry         | 0.486    | 0.436   | 0.231   | 0.201    |
> | History           | 0.420    | 0.441   | 0.367   | 0.430    |
> | Other             | 0.511    | 0.509   | 0.389   | 0.426    |
> | Health            | 0.526    | 0.562   | 0.480   | 0.502    |
> | Economics         | 0.646    | 0.655   | 0.582   | 0.576    |
> | Math              | 0.521    | 0.368   | 0.215   | 0.188    |
> | Physics           | 0.487    | 0.441   | 0.264   | 0.248    |
> | Computer Science  | 0.507    | 0.524   | 0.361   | 0.356    |
> | Philosophy        | 0.417    | 0.481   | 0.367   | 0.419    |
> | Engineering       | 0.355    | 0.372   | 0.226   | 0.214
> |||||
> |**Subject-Aggregate** | **0.500** | 0.492 | 0.377 | 0.379
> | **Overall Average** | **0.496** | 0.476  | 0.356 |0.353
>
> AZR **outperforms all three baselines**. Specifically, it achieves higher aggregate performance over **subjects** (0.500 for AZR vs. 0.492 for ORZ) and over all **data points** (0.496 for AZR vs. 0.476 for ORZ). These results indicate that AZR also demonstrates **strong general reasoning capabilities outside of math and code generation**.
>
> We will add these results to the updated manuscript and believe they greatly strengthen our findings.
>
> ---
> ## References
> [1] Zeng, Weihao, et al. "Simplerl-zoo: Investigating and taming zero reinforcement learning for open base models in the wild." arXiv preprint arXiv:2503.18892 (2025).
>
> [2] Wang, Yubo, et al. "Mmlu-pro: A more robust and challenging multi-task language understanding benchmark." Advances in Neural Information Processing Systems 37 (2024): 95266-95290.

---

> > ### Comment · Reviewer_Av4V · 2025-08-03
> >
> > Thanks for the authors' detailed response.
> >
> > I think the newly updated experimental results have addressed my concerns. I decided to raise my score.

---

> > > ### Author Response · Authors · 2025-08-04
> > >
> > > We would like to thank the reviewer for their time and effort throughout the entire review process. We believe that your suggestion to include general reasoning significantly strengthened our results. Thank you again for your thorough and thoughtful review.
> > >
> > > Best regards,
> > > The authors of “Absolute Zero: Reinforced Self-play Reasoning with Zero Data”

---

### Official Review · Reviewer_hGMY · 2025-07-01

**Clarity:** 4
**Significance:** 4
**Originality:** 4
**Rating:** 5
**Confidence:** 4

**Summary:**

Reinforcement learning with verifiable rewards has been successful at training long chain-of-thought "reasoning" models. However, the scalability of RLVR for reasoning is limited by (among other things) the availability of verifiable data. Typically, verifiable data is curated by humans. This paper attempts to remove the need for manual human curation + annotation effort in the RLVR for reasoning chain. The idea is to have a two-player game where a policy both proposes verifiable coding problems to be solved, and attempts to solve them. The policy is rewarded for both proposing learnable problems, and for solving the problems. The authors implement this two-player game and show that the model is able to self-improve by proposing solvable coding problems and solving them.

**Questions:**

I believe this paper deserves at least a solid accept, but have a couple of concerns that I would like the authors to answer before I raise my score.

1. You should cite the [Language Models Can Self-Improve at Writing Code](https://openreview.net/pdf?id=SaRj2ka1XZ3) paper in your Related Works, I feel it is a glaring omission given the similarities.
2. I don't understand how the Qwen-2.5-base if used as the proposer can follow such a complex prompt as given in Fig. 37. Please explain, or am I misunderstanding the setup? On another note, I find Table 1. kind of confusing, but it could just be my personal preference.

**Ethical Concerns:**

["NO or VERY MINOR ethics concerns only"]

**Final Justification:**

Using a simple, uncomplicated setup, the authors achieve a convincing demonstration of self-training (I do not like the word "self-improvement") using reinforcement learning without supervised finetuning data. I think this is an interesting result, especially given how _simple_ the method is. I think the community will appreciate it. Other reviewers are positive as well.

**Limitations:**

Yes

**Quality:**

4

**Strengths And Weaknesses:**

# Strengths
The main strength of the paper are the exciting application area (self-improvement) and the simplicity of the learning setup. The reward design for the proposer makes sense — the proposer should be rewarded for creating problems that are solvable, but not trivial or impossible. The reward design for the solver is likewise identical to the standard RLVR setup.

# Weaknesses
With respect to novelty, the idea that language models can self-improve at writing code by generating their own problems has already been shown by [Hapulztok et. al](https://openreview.net/pdf?id=SaRj2ka1XZ3). The key differences here are that the problem proposer is also trained in the work under review, and the experimental results are stronger. I don't see the use of RL as a big differentiator. I think enough time has passed and the context is different enough that the authors don't need to justify their novelty, but entirely missing it in the Related Works is not good.

I'm confused about the use of "base" model and the prompts. According to Fig. 37, the prompt for proposing problems is an elaborate prompt with multiple sections that ostensibly requires highly developed instruction following ability. However, training is started from the base models. I have a hard time believing this setup works — how is the base model able to follow such complex instructions?

---

> ### Author Rebuttal · Authors · 2025-07-25
>
> # Response
> We would like to thank the reviewer for their time, attention to detail, and thoughtful review. We address each of the weaknesses and questions below. We believe that **incorporating these responses and results into the updated manuscript has significantly improved its clarity and quality**. Please let us know if you have any further concerns or suggestions during the discussion period.
>
> ## Response to Weakness 1 and Question 1
> > W1: With respect to novelty, the idea that language models can self-improve at writing code by generating their own problems has already been shown by Hapulztok et. al...
>
> > Q1: You should cite...
>
> Thanks for bringing up the "Language Models Can Self-Improve at Writing Code" [1] paper. We genuinely missed this paper and acknowledge that it has many similarities with our work. We will **cite and include a detailed discussion** of this paper in the updated manuscript.
>
> Besides the differentiators mentioned by the reviewer, we would like to emphasize two more important differences with our work and Hapulztok et. al. or similar existing works:
> 1) Even though there are some existing self-improvement papers like the one mentioned by the reviewer, they are tested in the same domain as their training domain. While we also show improvements evaluated on **in-domain (code input/output/function prediction)** reasoning tasks (Figure 18), the finding that our method was able to generalize to **out-of-domain (math and code generation in the manuscript)** reasoning tasks (Table 1) and **outperform all other models** is perhaps even more important. Our designed tasks were crucial for developing generalized patterns, which are extremely important for out‑of‑distribution (OOD) performance (see the ablation in Table 2 of the manuscript). To further support our claim of superior OOD generalized reasoning, we **conducted a new experiment on general reasoning on MMLU-Pro**, comparing all baselines derived from Qwen2.5‑7B, including the best performing baseline ORZ-7b, all using greedy decoding and max output tokens=16k. We find that the AZR model still outperforms all baselines, achieving higher aggregate performance over subjects (0.500 for AZR vs. 0.492 for ORZ) and over all data points (0.496 for AZR vs. 0.476 for ORZ). These results further **strengthen our claim of improved OOD generalized reasoning**. See the table below this subsection for these results.
> 2) Furthermore, the reward for the AZR proposer forms an **automatic curriculum** for the learning policy, unlike other works that may generate tasks or goals more arbitrarily. Curriculum has been shown in the reasoning literature to both improve final performance [2] and be more data-efficient [3], and we were able to naturally incorporate it into our AZR method.
>
> |            | AZR_Base_7b | ORZ_7b     | Qwen_7b | SimpleRL_7b |
> |-------------------|----------|---------|---------|----------|
> | Business          | 0.558    | 0.440   | 0.347   | 0.261    |
> | Law               | 0.280    | 0.309   | 0.277   | 0.305    |
> | Psychology        | 0.608    | 0.630   | 0.541   | 0.551    |
> | Biology           | 0.679    | 0.717   | 0.626   | 0.626    |
> | Chemistry         | 0.486    | 0.436   | 0.231   | 0.201    |
> | History           | 0.420    | 0.441   | 0.367   | 0.430    |
> | Other             | 0.511    | 0.509   | 0.389   | 0.426    |
> | Health            | 0.526    | 0.562   | 0.480   | 0.502    |
> | Economics         | 0.646    | 0.655   | 0.582   | 0.576    |
> | Math              | 0.521    | 0.368   | 0.215   | 0.188    |
> | Physics           | 0.487    | 0.441   | 0.264   | 0.248    |
> | Computer Science  | 0.507    | 0.524   | 0.361   | 0.356    |
> | Philosophy        | 0.417    | 0.481   | 0.367   | 0.419    |
> | Engineering       | 0.355    | 0.372   | 0.226   | 0.214
> |||||
> |**Subject-Aggregate** | **0.500** | 0.492 | 0.377 | 0.379
> | **Overall Average** | **0.496** | 0.476  | 0.356 |0.353
>
> ## Response to Weakness 2 and Question 2
> > W2: I'm confused about the use of "base" model and the prompts... how is the base model able to follow such complex instructions?
>
> > Q2: I don't understand how the Qwen-2.5-base if used as the proposer can follow such a complex prompt as given in Fig. 37...
>
> You are understanding the setup correctly, we did use the prompt in Fig.37 with the base model. We agree that the instructions seem complicated for base models, but both Qwen-2.5-Base and Llama-3.1-Base were sometimes able to produce desired results. We did indeed observe that they both had **difficulty with instruction-following** most of the time in the beginning. However, since the proposer step is also trained using RL, the formatting and task intent are gradually instilled into the model through RL training; concretely, **valid formatting for proposer** at step 0 was 7% and at step 60 it reaches over 90%. Therefore, we do not need the model to perfectly follow the complex instructions in the beginning and just a handful of successful outputs at first can allow the model to **gradually learn how to reliably generate the required task outputs**.
>
> One conjecture as to why some models, like Qwen-2.5's base model, are able to sometimes follow more complex instructions out of the box is that they have some synthetic instruction data in the pretraining data mix. See "Qwen2.5 Technical Report" [4], Section 3.1.
>
>
> ### Response to Confusion about Table 1
> Do you mind sharing what aspect of Table 1 is difficult to understand? We are glad to incorporate your suggestions in the updated version of Table 1 in the revised manuscript. Thank you.
>
> ---
> ## References
> [1] Huang, Jiaxin, et al. "Large language models can self-improve." arXiv preprint arXiv:2210.11610 (2022).
>
> [2] Chen, Xiaoyin, et al. "Self-Evolving Curriculum for LLM Reasoning." arXiv preprint arXiv:2505.14970 (2025).
>
> [3] Yu, Qiying, et al. "Dapo: An open-source llm reinforcement learning system at scale." arXiv preprint arXiv:2503.14476 (2025).
>
> [4] Yang, Qwen An et al. "Qwen2.5 Technical Report." arXiv preprint arXiv:2412.15115 (2024).

---

> ### Author Response · Authors · 2025-08-04
>
> Dear Reviewer hGMY,
>
> We would like to once again thank you for your time and effort in providing a thoughtful review of our manuscript.
>
> First, we have included the missing citation you mentioned.
>
> Additionally, we would like to provide more context regarding points `W2` and `Q2`, which relate to the **instruction-following abilities of current base models**. To clarify, the base models (Qwen2.5 and Llama3.1) did not perfectly follow instructions from the outset, but were able to produce properly formatted proposer responses only a small fraction of the time, approximately one out of sixteen for Qwen2.5 and one out of 128 for Llama3.1. However, since the proposer is trained using a format and learnability reward, proper formatting and instruction following are gradually learned through reinforcement learning. After about 60 steps, the proposer is able to output properly formatted responses around 90 percent of the time. In summary, while the base model initially struggles with perfectly following (/understanding) complex instructions, it produces enough correctly formatted samples for reinforcement learning to be effective after multiple attempts.
>
> We also added more results regarding general reasoning (MMLU-Pro) outside of math/code generation and again was able to outperform baselines in aggregate.
>
> We believe these points comprehensively address your concerns. If you have any additional concerns, please feel free to let us know. Since you stated that our manuscript `deserves at least a solid accept`, we would be grateful if you could consider raising our score. Thank you again for your time and effort.

---

> > ### Comment · Reviewer_hGMY · 2025-08-05
> >
> > Thank you for the additional information. The additional details regarding the base models success rate over time is very interesting (at least to me) and I encourage you to add the results to the supplementary material or generally discuss it further, I think the notion that base models can be used in this way is surprising will be surprising to a lot of reviewers. I have raised my score. Best of luck!

---

> > > ### Author Response · Authors · 2025-08-06
> > >
> > > We are thankful for the reviewer’s time and effort throughout the entire process. We believe that the reviewer’s point about the success rate of proposer's formatting should indeed be highlighted in the paper. We will include it in the updated manuscript accordingly. Thank you again for your support and positive evaluation of our work.

---

### Official Review · Reviewer_oKgA · 2025-07-02

**Clarity:** 3
**Significance:** 3
**Originality:** 3
**Rating:** 5
**Confidence:** 4

**Summary:**

This paper first introduces a new RL setting “Absolute Zero”: RL from base models (zero setting) and with absolutely no supervised data (zero data). It also introduces a new pipeline to train a model in such a zero setting. The pipeline involves training a propser and a sovler, with a novel objective.

**Questions:**

1. What do you think are the necessary properties for proposed pipeline to be effective? Absolute Zero doesn’t work so well on Llama, could you explain more in depth?
2. What is the entropy dynamics during the training? Will the training also suffer from entropy collapse that limits further gain?
3. What will the results of pass@k be like? Will the absolute zero training be able to improve pass@k as well or are we trading pass@k for pass@1?

**Ethical Concerns:**

["NO or VERY MINOR ethics concerns only"]

**Final Justification:**

I believe this work is quite significant, as there is growing interest in this area of self-evolving agent.

The paper is generally well done, with extensive experiments, though I don't like the way the authors are pressuring for higher rating.

**Limitations:**

yes

**Quality:**

3

**Strengths And Weaknesses:**

**Strength:**
1. The paper is well written and articulate the motivation (the need for absolute zero RL to reach super human intelligence) well. The paper is also easy to follow.
2. The paper is generally novel. They were concurrent work in the “weak-supervised” RL setting. This paper has a unique and practical implementation in this direction.
3. Results on Qwen models are relatively strong. The author claims that there are better than baselines like SimpleRL.

**Weakness:**
1. The paper has the motivation of super-intelligence alignment. However, there is still a gap between the scale of current experiments and that vision. They were stronger RL trained model at the scale. I understand the computational constraints, but am still a bit reserved regarding that.
2. The results report greedy decoding. One may suspect that the training leads to decrease in pass@k and that the Absolute Zero training does not broaden the capability of the model. One may suspect that the absolute zero training is exhausting the potential of the base model.

---

> ### Author Rebuttal · Authors · 2025-07-29
>
> # Response
> We would like to thank the reviewer for their feedback, particularly their concerns focusing on convergence properties and scaling. We address each of the weaknesses and questions below. By incorporating these responses and results, **particularly the reviewer’s suggestion that led to our new finding that AZR training performs strongly even at high pass@512 compared to the base model in math and coding**, will significantly enhance the strength of our claims and clarity of our paper. We look forward to the discussion period to further engage, especially to clarify a point the reviewer raised in **Weakness 1**. Thank you.
>
> ## Response to Weakness 1
> > W1: The paper has the motivation of super-intelligence alignment...
>
> Thank you for the feedback. We believe a critical aspect of progress toward improving superhuman intelligence is a **model’s ability to self-evolve without human intervention**. In our manuscript, we demonstrate AZR can self-evolve by proposing its own tasks and learning from them with access to only Python, even surpassing models that have access to expert human data. We believe this is a meaningful demonstration of a viable self-improving reasoning system, and that its importance is independent of scale.
>
> In our experiments we also observed promising scaling properties. Specifically, when we increased the model size from 3b to 7b and to 14b, the **percentage gain in general reasoning performance grew at each scale** (Fig. 3b). This pattern suggests that our training framework scales favorably, a full investigation in its scaling law is promising future work.
>
> Lastly, could you **please clarify what you meant by** "They were stronger RL-trained models at the scale"? Thank you.
>
> ## Response to Weakness 2 and Question 3
> > W2: ... One may suspect that the training leads to decrease in pass@k and that the Absolute Zero training does not broaden the capability of the model...
>
> > Q3: What will the results of pass@k be like? ...
>
> Great question. We were also eager to evaluate the **reasoning coverage** of our model using different pass@k's. We follow the protocol from Yang et al. \[1], using temperature = 0.6, top\_p = 0.95, max output tokens=16k, and a maximum high *k* of 512 (most results from Yang et al. presented a lower *k* of 128). Since we cannot provide visual plots, we aid interpretation by including a final **Δ** row that shows the pass\@k difference between the AZR model and the base model: $Δ = Pass@k_{AZR} - Pass@k_{Base}$ **(positive Δ means AZR is outperforming the base model)**. We present results on three **code generation benchmarks** (LiveCodeBench, MBPP++ and HumanEval++) and two **math benchmarks** (AIME24 and AIME25). Note that LiveCodeBench goes up to pass@256 due to overwhelming run time (days).
>
> ### **MBPP++**
>
> |            | pass@1    | pass@2    | pass@4    | pass@8    | pass@16   | pass@32   | pass@64   | pass@128  | pass@256  | pass@512  |
> |:----------|:------:|:------:|:------:|:------:|:-------:|:-------:|:-------:|:--------:|:--------:|:--------:|
> | AZR-Base-7b        | 66.6 | 73.2 | 77.9 | 81.4 | 83.9 | 85.6 | 87.1 | 88.4 | 89.7 | 91.0 |
> | Qwen2.5 7b | 63.2 | 70.4 | 75.6 | 79.5 | 82.4 | 84.4 | 85.7 | 86.7 | 87.2 | 87.6 |
> | Δ          | +3.4 | +2.8 | +2.3 | +1.9 | +1.5 | +1.2 | +1.4 | +1.7 | +2.5 | +3.4 |
>
> ---
>
> ### **Humaneval++**
>
> |            | pass@1    | pass@2    | pass@4    | pass@8    | pass@16   | pass@32   | pass@64   | pass@128  | pass@256  | pass@512  |
> |:----------|:------:|:------:|:------:|:------:|:-------:|:-------:|:-------:|:--------:|:--------:|:--------:|
> | AZR-Base-7b | 69.9 | 77.2 | 82.0 | 85.9 | 89.1 | 91.3 | 93.0 | 94.3 | 95.4 | 95.7 |
> | Qwen2.5 7b | 68.5 | 76.8 | 81.8 | 84.8 | 86.9 | 88.8 | 90.4 | 91.8 | 93.1 | 94.5 |
> | Δ          | +1.4 | +0.4 | +0.2 | +1.1 | +2.2 | +2.5 | +2.6 | +2.5 | +2.3 | +1.2 |
>
> ---
>
> ### **LiveCodeBench**
>
> |            | pass@1    | pass@2    | pass@4    | pass@8    | pass@16   | pass@32   | pass@64   | pass@128  | pass@256  |
> |:----------|:------:|:------:|:------:|:------:|:-------:|:-------:|:-------:|:--------:|:--------:|
> | AZR-Base-7b| 24.8 | 30.7 | 36.1 | 41.0 | 45.3 | 49.3 | 52.8 | 55.9 | 58.9 |
> | Qwen2.5 7b | 20.2 | 25.7 | 30.7 | 35.3 | 39.3 | 42.9 | 46.0 | 48.6 | 50.9 |
> | Δ          | +4.6 | +5.0 | +5.4 | +5.7 | +6.0 | +6.4 | +6.8 | +7.3 | +8.0 |
>
> ---
>
> ### **AIME25**
>
> |            | pass@1    | pass@2    | pass@4    | pass@8    | pass@16   | pass@32   | pass@64   | pass@128  | pass@256  | pass@512  |
> |:----------|:------:|:------:|:------:|:------:|:-------:|:-------:|:-------:|:--------:|:--------:|:--------:|
> | AZR-Base-7b| 7.0  | 11.8 | 17.8 | 23.7 | 29.1 | 35.0 | 41.3 | 47.1 | 52.7 | 60.0 |
> | Qwen2.5 7b | 3.5  | 6.3  | 10.6 | 16.2 | 22.7 | 29.8 | 36.8 | 43.3 | 48.9 | 53.3 |
> | Δ          | +3.5 | +5.5 | +7.2 | +7.5 | +6.4 | +5.2 | +4.5 | +3.8 | +3.8 | +6.7 |
>
> ---
>
> ### **AIME24**
>
> |            | pass@1    | pass@2    | pass@4    | pass@8    | pass@16   | pass@32   | pass@64   | pass@128  | pass@256  | pass@512  |
> |:----------|:------:|:------:|:------:|:------:|:-------:|:-------:|:-------:|:--------:|:--------:|:--------:|
> | AZR-Base-7b| 12.4 | 16.8 | 21.1 | 25.6 | 31.1 | 37.8 | 45.2 | 52.6 | 58.6 | 63.3 |
> | Qwen2.5 7b | 7.0  | 11.2 | 16.4 | 22.5 | 29.0 | 35.5 | 42.1 | 48.6 | 55.8 | 66.7 |
> | Δ          | +5.4 | +5.6 | +4.7 | +3.1 | +2.1 | +2.3 | +3.1 | +4.0 | +2.8 | –3.4 |
>
> At very high pass@k (256 and 512), **the AZR model still consistently outperforms the base model across all five benchmarks** (exception AIME24 pass@512). Compared to results from Yang et al. [1], we find that **AZR-trained models maintain strong gains even at high pass@k**, highlighting a new and notable finding. This implies that **our models maintained healthy reasoning coverage and answer entropy even after RL training**, which is *beneficial for applying further test-time scaling or search methods that rely on generating diverse samples*. We believe this finding substantially strengthens our results and will feature it in the updated manuscript, along with a detailed discussion. Thank you for your suggestion.
>
> ## Response to Question 1
> > Q1: What do you think are the necessary properties for proposed pipeline to be effective?...
>
> Based on the comprehensive experiments we conducted for this work, we believe that base capability is important for more improvements with AZR training, and we found that base capability has a **compounding effect on the final performance gain**. This again ties to the reviewer’s point in [Response to Weakness 1](#response-to-weakness-1), where we indeed saw scaling to be favorable for AZR, with 3b, 7b, and 14b models showing a higher increase in final performance after AZR training as we scale bigger. We see this as a strength of our method because models in our field are constantly becoming more capable.
>
> As for why the Llama model does not achieve strong improvements, it is due to the weaker base model capabilities mentioned above, which have also been observed in prior works, for example, the lack of reflection patterns in its data mix [1, 2, 3]. Since AZR uses online RL to train its models, we encounter similar issues. Additionally, because our framework proposes its own tasks, using a weaker model as the base may introduce a bottleneck in generating tasks that are meaningful and learnable. We believe that supervised fine-tuning on strong mid-stage data, as done in [1, 2], can help Llama benefit more significantly from AZR training afterwards. We view this as an orthogonal but promising direction for future work.
>
> ## Response to Question 2
> > Q2: What is the entropy dynamics during the training? Will the training also suffer from entropy collapse that limits further gain?
>
> Thank you for the question. We observed different patterns of entropy during training. In some runs, the entropy hovered at around 0.2, while in others it gradually decreased. However, we did not see much generalized reasoning performance differences or correlations between different patterns.
>
> One particularly interesting observation is that although our AZR 7B model snapshot has a mean token-level entropy of around 0.2, which is lower than the base model’s 1.0; however, our additional pass\@k experiments, in [Response to Weakness 2 and Question 3](#response-to-weakness-2-and-question-3), show that AZR **consistently improves from pass\@1 to pass\@k** and fully covers the corresponding range achieved by the base model. AZR model's consistent improvement from pass\@1 to pass\@k indicates that the model is still exploring **diverse reasoning trajectories** that lead to correct answers. In other words, while token-level entropy is lower, the **entropy among final answers [4, 5], or the semantic entropy [6] of reasoning traces, remains healthy** because multiple valid solutions continue to be produced across different samples. We believe this form of entropy is a more suitable measure of whether a policy has truly "collapsed".
>
> ---
> ## References
> [1] Gandhi, Kanishk, et al. "Cognitive behaviors that enable self-improving reasoners, or, four habits of highly effective stars." arXiv preprint arXiv:2503.01307 (2025).
>
> [2] Wang, Zengzhi, et al. "Octothinker: Mid-training incentivizes reinforcement learning scaling." arXiv preprint arXiv:2506.20512 (2025).
>
> [3] Zeng, Weihao, et al. "Simplerl-zoo: Investigating and taming zero reinforcement learning for open base models in the wild." arXiv preprint arXiv:2503.18892 (2025).
>
> [4] Lyu, Qing, et al. "Calibrating large language models with sample consistency." Proceedings of the AAAI Conference on Artificial Intelligence. Vol. 39. No. 18. 2025.
>
> [5] Zhang, Mozhi, et al. "Calibrating the confidence of large language models by eliciting fidelity." arXiv preprint arXiv:2404.02655 (2024).
>
> [6] Kuhn, Lorenz, Yarin Gal, and Sebastian Farquhar. "Semantic uncertainty: Linguistic invariances for uncertainty estimation in natural language generation." arXiv preprint arXiv:2302.09664 (2023).

---

> ### Author Response · Authors · 2025-08-04
> **Rebuttal Summary**
>
> Dear Reviewer oKgA,
>
> We would like to thank you for your time and effort in reviewing our work. Below you can find a concise summary of our two most important responses during the rebuttal period:
>
> ---
> 1. We provided **new results showing that a high pass\@k was able to outperform the base model, even at values as high as 512, on math and code generation benchmarks**. This highlights that our model is complementary to further test-time scaling methods that require answer diversity, and our model retained healthy answer/CoT entropy throughout training.
> 2. Our **AZR framework favors scaling, and better initial base capabilities serve as a catalyst for subsequent improvements**. As models generally improve over time, we expect our framework to benefit increasingly from more capable models.
> ---
>
> We believe these two points address the main concerns raised by the reviewer. Also, could you please clarify what you meant by "They were stronger RL-trained models at the scale"? If you have any further questions, please let us know so we can provide additional information. If we have comprehensively addressed your concerns, we would be grateful if you can consider raising our score. Thank you.

---

> > ### Comment · Reviewer_oKgA · 2025-08-05
> >
> > Thank you for the additional results on pass@k. I am not very sure about the decoding settings though. From my own experiments, around 10% of delta can be explained by changes of decoding settings like temperature. I do understand that conducting temperature sweeps is very expensive.
> >
> > Anyway I think this work is significant as there is growing interest in self-play agent.
> >
> > I will update my score accordingly.

---

> > > ### Author Response · Authors · 2025-08-06
> > >
> > > Thank you for acknowledging our work as significant.
> > >
> > > For the pass\@k results, we **strictly followed all the settings from Yang et al. \[1]** (the main paper that proposed the pass\@k metric for reasoning), using temperature = 0.6 and top\_p = 0.95, which are **widely adopted decoding settings in many recent works** \[2, 3, 4, 5]. Therefore, we believe our decoding setting is fair and produces a significant result. Furthermore, we agree that the bounded reasoning coverage issue in RLVR, as demonstrated by Yang et al. \[1] (a concurrent work of ours), is important and remains an ongoing challenge that is actively being investigated \[6]. However, we believe it is an orthogonal direction to our manuscript's objectives.
> > >
> > > Our main contribution is to demonstrate that **without human-curated data, LLM reasoners can learn to self-evolve by proposing relevant tasks and outperforming models trained with human-curated data under similar model size, RL algorithm and data scales**. Furthermore, **new results from our rebuttal show that AZR is strong at pass\@k and achieves high reasoning coverage compared to the base model**, which we believe effectively addresses the reviewer's original points (W2, Q2, Q3) about potential collapse to only pass\@1 performance.
> > >
> > > We hope our response helped further elucidate our objectives and our entire rebuttal fully addressed the reviewers' concerns. Additionally, we noticed that the reviewer mentioned they would update the score, but it was not changed, we kindly ask if the reviewer forgot to do so. We hope this response encourages the reviewer to reconsider and, hopefully, raise our score. Thank you again for your engagement, time, and thoughtful comments throughout the review process.
> > >
> > > ---
> > > [1] Yue, Yang, et al. "Does reinforcement learning really incentivize reasoning capacity in llms beyond the base model?." arXiv preprint arXiv:2504.13837 (2025).
> > >
> > > [2] Liu, Mingjie, et al. "Scaling Up RL: Unlocking Diverse Reasoning in LLMs via Prolonged Training." arXiv preprint arXiv:2507.12507 (2025).
> > >
> > > [3] Cheng, Daixuan, et al. "Reasoning with exploration: An entropy perspective." arXiv preprint arXiv:2506.14758 (2025).
> > >
> > > [4] Liu, Mingjie, et al. "Prorl: Prolonged reinforcement learning expands reasoning boundaries in large language models." arXiv preprint arXiv:2505.24864 (2025).
> > >
> > > [5] Chen, Yang, et al. "Acereason-nemotron: Advancing math and code reasoning through reinforcement learning." arXiv preprint arXiv:2505.16400 (2025).
> > >
> > > [6] Wu, Fang, et al. "The Invisible Leash: Why RLVR May Not Escape Its Origin." arXiv preprint arXiv:2507.14843 (2025).

---

> > > > ### Author Response · Authors · 2025-08-08
> > > >
> > > > As we have less than 24 hours left in the rebuttal, we would be grateful if the reviewer can express whether our last response addressed their final concerns. Thanks to the reviewers point on pass@k, we believe adding these results to our final manuscript will really strengthen our results. If the reviewer believe our efforts and responses are sufficient, we kindly ask if you could revise your final score. We appreciate your time and efforts throughout the whole reviewing process. We are still on standby for any remaining questions/concerns.

---

> > > > > ### Comment · Reviewer_oKgA · 2025-08-09
> > > > >
> > > > > Dear authors,
> > > > >
> > > > > Don't panic. I was just adhering to the suggested timeline of posting final rating after discussion period.

---

### Decision · Program_Chairs · 2025-09-17

**Decision:**

Accept (spotlight)

**Comment:**

The reviewers are overall positive and converge on recommending acceptance. They highlight the novelty and simplicity of the “Absolute Zero” framework, which enables self-play reinforcement learning for reasoning without any supervised or human-curated data. Strengths include a clear motivation, a practical implementation, and strong experimental results across code and math reasoning benchmarks, with convincing demonstrations of generalization to out-of-domain tasks. Reviewers noted that the method scales favorably, produces harder tasks with larger models, and maintains strong performance even at high pass@k, alleviating concerns about entropy collapse or overfitting to pass@1. The work is also recognized as significant in light of the growing interest in self-evolving reasoning agents.

Concerns raised included questions of scalability to super-intelligence alignment (which motivated the paper), whether improvements extend beyond greedy decoding to broader coverage (pass@k), the complexity of generated tasks, and a missing related-work citation. These were substantively addressed in the rebuttal with additional experiments (e.g., pass@512 performance, MMLU-Pro results, frontier model difficulty evaluations) and clarifications. Reviewers expressed satisfaction with these additions, and two explicitly raised their scores. No major disagreements remain, though one reviewer noted some reservations about long-term convergence and scale, which do not undermine the paper’s contributions.